

# Land surface model parameter optimisation using in-situ flux data: comparison of gradient-based versus random search algorithms

Vladislav Bastrikov[1,2], Natasha MacBean[1*], Cédric Bacour[3], Diego Santaren[1], Sylvain Kuppel[4] and Philippe Peylin[1]

[1]Laboratoire des Sciences du Climat et de l'Environnement, UMR 8212 CEA-CNRS-UVSQ, 91191 Gif-sur-Yvette, France
[2]Institute of Industrial Ecology UB RAS, 620219 Ekaterinburg, Russia
[3]Noveltis, 153 rue du Lac, 31670 Labège, France
[4]Northern Rivers Institute, University of Aberdeen, Aberdeen, AB24 3UF, United Kingdom
*Now at School of Natural Resources and the Environment, University of Arizona, 1064 E Lowell St., 85721, Tucson, AZ, USA

*Correspondence to*: Vladislav Bastrikov (vladislav.bastrikov@lsce.ipsl.fr)

**Abstract.** Land surface models (LSMs), used within earth system models, rely on numerous processes for describing carbon, water and energy budgets, often associated with highly uncertain parameters. Data assimilation (DA) is a useful approach for optimising the most critical parameters in order to improve model accuracy and refine future climate predictions. In this study, we compare two different DA methods for optimising the parameters of seven plant functional types (PFTs) of the ORCHIDEE land surface model using daily averaged eddy-covariance observations of net ecosystem exchange and latent heat flux at 78 sites across the globe. We perform a technical investigation of two classes of minimisation methods — local gradient-based (the L-BFGS-B algorithm) and global random search (the genetic algorithm) — by evaluating their relative performance in terms of the model–data fit and the difference in retrieved parameter values. We examine the performance of each method for two cases: when optimising parameters at each site independently ("single-site" approach) and when simultaneously optimising the model at all sites for a given PFT using a common set of parameters ("multi-site" approach). We find that for the single site case the random search algorithm results in lower values of the cost function (i.e. lower model – data root mean square differences) than the gradient-based method; the difference between the two methods is smaller for the multi-site optimisation due to a smoothing of the cost function shape with a greater number of observations. The spread of the cost function, when performing the same tests with 16 random first guess parameters, is much larger with the gradient based method, due to the higher likelihood of being trap in local minima. When using pseudo-observations tests the genetic algorithm results in a closer approximation of the true posterior parameter value in the L-BFGS-B algorithm. We demonstrate the advantages and challenges of different DA techniques and provide some advice on using it for the LSM parameters optimisation.



## 1 Introduction

In the context of climate change and the potentially large impact of global warming on terrestrial ecosystem functioning (and vice versa), the development of reliable and robust process-driven land surface models (LSMs) to assess the future evolution of carbon, water and energy budgets is of key importance (Field et al., 2004). These models also represent a major tool for

understanding the underlying mechanics and interconnections between the terrestrial biosphere, the atmospheric composition and human environmental management practices (Cramer et al., 2001).

State of the art LSMs include hundreds of processes describing energy and mass (water and carbon) transfers in the soil – plant – atmosphere continuum on a wide range of temporal and spatial scales. This inevitably leads to high model complexity due to interactions between biophysical and biogeochemical processes. The simulated carbon, water and energy

fluxes and stocks thus remain highly uncertain (Anav et al., 2013; Friedlingstein et al., 2014; Sitch et al., 2015). These uncertainties typically arise from three main sources:

- errors in the input data: uncertainty of the meteorological forcing and/or the vegetation and soil spatial information used to drive the model;
- errors in the model structure: insufficient or erroneous description of biogeochemical and biophysical processes;

- errors in the model parameterisation: uncertain or incorrect values of the fixed values (parameters) in the model.

The latter source of uncertainty is the focus of this study. Model parameter error can be due to insufficient availability or inadequacy of observations and experiments used to derive the model parameter values, or because the parameters themselves are "effective" and not directly observable in reality. In the former case, observations were usually limited to only a few experiments carried out at spatial/temporal scales that were not always relevant for larger scale simulations. For

instance, the observations correspond to experiments using only a few plant species, soil types or climatic regions, and thus typically did not encompass all the variability required of the vegetation classification used in LSMs based on the concept of plant functional types (PFTs). Parameter uncertainty can therefore lead to significant uncertainty in simulations of carbon, water and energy fluxes and stocks (Luo et al., 2016; Peylin et al., 2016; Raoult et al., 2016; Schürmann et al., 2016; Thum et al., 2017).

Fortunately, there are a growing number of observations as well as new types of measurements that should aid us in significantly reducing model parameter uncertainty. Among these observations, eddy-covariance measurements of net CO2 flux (net ecosystem exchange – NEE), water and energy fluxes at specific FLUXNET sites represent an unprecedented source of information on the diurnal to seasonal and inter-annual processes (Williams et al., 2009). More than 650 sites, operating on a long-term and continuous basis and covering major ecosystem of the world, are available (Baldocchi et al.,

2001; http://fluxnet.fluxdata.org). A large number of studies have used these flux measurements to optimise ecosystem model parameters, either using the data from one single site (SS) (Reichstein et al., 2003; Braswell et al., 2005; Moore et al., 2008; Ricciuto et al., 2011; Santaren et al., 2014; Kato et al., 2015; Bacour et al., 2015) or from multiple sites (MS) simultaneously usually at the level of PFT (Groenendijk et al., 2011; Kuppel et al., 2014; Raoult et al., 2016). These studies



have employed various data assimilation (DA) techniques, but they all rely on a Bayesian framework that allows the update of a priori information on the parameters (i.e., the prior probably distribution functions – PDFs) with information contained in the observations, leading to the posterior PDFs. While some methods do not require any assumption as to the shape of the different PDFs (see for example Van Oijen et al., 2005) most of them use Gaussian error assumptions (for both parameters and observations), which simplifies the characterisation of the posterior parameter PDFs and leads to the minimisation of a cost function (Dietze et al., 2013). Our study uses such Gaussian hypothesis.

The optimisation requires the minimisation of a cost function that describes the difference between the model and the observations, taking into account their uncertainties. There is a wide variety of mathematical algorithms that can be used to locate the minimum of a cost function and these methods can be broadly divided into two classes, i) local gradient-based method (i.e., gradient-descent or conjugate gradient algorithms for example) also referred as variational method and ii) global random search method (i.e., Markov Chain Monte Carlo (MCMC), genetic and particle filter algorithms for the most common ones). There are various advantages and disadvantages for the different algorithms of each class, depending on the model complexity (mainly the degree of model non-linearity), the simulation computational time and the number of parameters to be optimised. Random search methods have been proven to be efficient in finding the correct parameter values with highly non-linear models but at the expense of potentially prohibitive computing cost. On the other hand, gradient-based methods are more prone to getting stuck in local minima and, furthermore, they require the complex calculation of the gradient of the cost function with respect to all parameters. With idealised models, Chorin and Morzfeld (2013) have shown that the assimilation can be optimal with particle filters or variational methods, under the condition that the dimension of the problem is not excessive. The choice of the minimisation method was shown to have little impact on the overall optimisation efficiency for relatively simple ecosystem models (Trudinger et al., 2007; Fox et al., 2009). Ziehn et al. (2012) showed similar performances between a MCMC and a variational method in terms of locating the minimum of the cost function for the BETHY ecosystem model (using atmospheric $CO_2$ data as constraint), except for the difference in computational time (much longer using MCMC). However, with the more complex ORCHIDEE LSM, Santaren et al. (2014) showed that the genetic algorithm was superior to a gradient-based method in reducing the cost-function to the correct minimum at one FLUXNET site (using water and carbon fluxes as constraint), as the gradient method was often stuck in local minima. Our study further expands their analysis to multiple-sites and across multiple PFTs. The overall objective of this study therefore is to investigate the trade-off between these two classes of method in terms of their computational efficiency versus their ability to constrain the parameters to the correct value. To achieve this goal, we performed experiments using i) a global state of the art process-based land surface model (ORCHIDEE, Krinner et al. 2005), ii) a large ensemble of $CO_2$ and water flux measurement at FluxNet sites and iii) one particular variant for each class of method, the gradient-based L-BFGS-B algorithm (Byrd et al., 1995) and a genetic algorithm (GA) for the random search method. The key questions that we investigate are the following:

   i)   Does the random search algorithm (GA) result in a lower spread in cost function minimum than the L-BFGS_B gradient-based method (BFGS) for the single site (SS) tests?



ii) Are the differences in cost function minimum between the GA and BFGS methods smaller for multi-site (MS) optimisations than for single sites (SS), following a potential smoothing of the cost function shape with a greater number of observations?

iii) What is the spread of the posterior parameter values with the BFGS and the GA methods when performing the same tests with multiple first guesses in parameter space? Is the spread different between the two methods and how many first guesses are needed to improve the location of the cost function minimum?

iv) Does one method result in a closer approximation of the true posterior parameter value in the pseudo-observation tests, for both the SS and MS experiments?

Section 2 presents the ORCHIDEE model and its configuration, describes the observational data exploited in the study and gives a detailed explanation of the two data assimilation methods as well as the setup of the experiments. A comparison of the performances of the two optimisation methods and of the differences between the single and multiple sites approaches is done in Section 3, including the analysis of model–data misfit (Sect. 3.1) and estimated parameter values (Sect. 3.2). The last section summarises the main findings and proposes few perspectives to this work.

## 2 Materials and methods

### 2.1 The ORCHIDEE land surface model

We used the process-based land surface model ORCHIDEE (ORganizing Carbon and Hydrology In Dynamic Ecosystems) that has been developed at the IPSL (Institut Pierre Simon Laplace, France). It simulates the carbon, water and energy exchanges between the land surface and the atmosphere (Krinner et al., 2005). The model is applicable over a wide range of spatial scales (from "grid-point" to regional or global) and covers timescales from 30 minutes to possibly thousands of years. ORCHIDEE may be deployed as a stand-alone terrestrial biosphere model using meteorological forcing (observational data or model reanalysis) or as part of the IPSL Earth System Model (Dufresne et al., 2013) used for climate predictions. The version that we use corresponds to the one used for the CMIP5 simulations for the IPCC 5th Assessment Report.

The hydrological and photosynthesis processes as well as the energy balance are treated at a half-hourly time step, while the slower components linked to the carbon allocation within the plants, the autotrophic respiration, leaf onset and senescence, plant mortality and soil organic matter decomposition are processed at a daily time step. The soil hydrology model used in this study corresponds to a simple double-bucket scheme (Ducoudré et al., 1993). The land surface is represented by 13 plant functional types (PFT), including bare soil, that can co-exist in any grid cell. Except for the phenology, the processes are described by generic equations but with different parameters that are PFT-specific. For the main model equations, the reader is referred to numerous previous publications (e.g. Krinner et al., 2005) that are reported on the ORCHIDEE web-site (http://orchidee.ipsl.fr).

For this study we applied the model at site level (see section 2.3) using gap-filled half-hourly meteorological data measured at each site (air temperature, humidity, pressure, wind speed, rainfall and snowfall rates, shortwave and longwave incoming



radiation; see Vuichard et al. 2015). The soil carbon pools have been brought to equilibrium (spin-up procedure) by cycling the available meteorological forcing over several millennia (with present day $CO_2$ concentration) in order to ensure that the long-term net carbon flux is close to zero.

## 2.2 Optimised parameters

The ORCHIDEE parameters selected for the optimisation are described in Table 1 along with their default values. They have been chosen based on the sensitivity of the simulated daily net $CO_2$ ecosystem exchange (NEE) and latent heat fluxes (LE) to all related model parameters (not shown). Overall, the 28 most sensitive parameters were selected, as in recent optimisation studies with ORCHIDEE (e.g. Santaren et al., 2014 and Kuppel et al., 2014). The parameters control photosynthesis, respiration fluxes, leaf phenology, plant water stress and energy balance. 16 parameters are PFT-dependent, 11 parameters are PFT-independent and 1 parameter ($K_{soilC}$) is site-specific, which results in a total number of 192 parameters. $K_{soilC}$ scales the initial values (after spin up) of the slow and passive soil carbon pools in order to account for the historical effects (e.g. land use change) that are not accounted for in the model spin-up that would result in a disequilibrium of these pools in reality (e.g. changes in land use and management). The parameter bounds and uncertainties used in this study were derived from expert knowledge of the model equations; they are reported in the Supplement (Table S1). The prior uncertainty is set to 40% of the range of variation for each parameter, following previous studies (Kuppel et al., 2012; MacBean et al., 2015).

## 2.3 Assimilated data: carbon and water fluxes

We use eddy covariance measurements of net surface $CO_2$ fluxes (referred as NEE) and latent heat flux (LE) from the FLUXNET global network (La Thuile database, Baldocchi, 2001). We selected 78 sites, representing 7 PFTs of ORCHIDEE (from the total number of 13) with the number of sites in each PFT varying from 2 (temperate evergreen broadleaf forests (TempEBF)) to 24 (C3 grasses) and the length of each observation record varying from 1 to 10 years. We only kept sites where the main PFT represents at least 70% of the footprint of the flux tower so that we could restrict the optimisation to the parameters of one PFT per site. We disregarded sites that had a large discrepancy with respect to the prior ORCHIDEE simulation (in terms of seasonal cycle) which suggests large model structural errors that prevent meaningful parameter optimisations. Finally, we only use continuous flux time series that do not contain gaps of more than few days. This subset of FLUXNET sites corresponds to the one used by Kuppel et al. (2014). The selected sites (geographical location, data period and references) are described in the Supplement (Table S2). Where possible, the LE fluxes have been corrected to close the energy balance, using measurements of the ground heat flux ($G$) and keeping a constant Bowen ratio (to update the latent and sensible heat fluxes; Twine et al., 2000). Individual days with less than 80% of data coverage have not been used.

For the assimilation, the half-hourly observations have been averaged on a daily basis and smoothed with a 15-days running mean, following Bacour et al. (2015). This allows us to focus the optimisation on the seasonal and annual time scales, excluding the influence of short-term flux variations. In order to test the influence of smoothing the data on the optimisation performance, preliminary assimilation experiments were run with different smoothing configurations (10-days running



mean, 5-days running mean and without smoothing). Overall the choice of the smoothing configuration had little impact on the outcome of this study. A slight decrease of the ability to find the minimum is observed with the gradient-based method when narrowing the smoothing interval for the NEE/LE fluxes. With the random search algorithm the impact is even smaller. The reader is referred to the Supplement for the illustration of the smoothing influence on the fit to the data (Figure

S1, example for TempDBF PFT, last columns). From the results of this test we determined that the key messages of the results are not impacted by the smoothing window; therefore, the results are based on the 15-day running mean.

## 2.4 Data assimilation framework

### 2.4.1 The principle of minimisation

The optimisation methodology used in this study relies on a Bayesian statistical formalism (see for instance Tarantola, 2005,
chapter 3). Assuming that the errors associated with the parameters, the observations and the model outputs follow Gaussian distributions, the optimal parameter set corresponds to the minimum of a cost function, $J(x)$, that measures the mismatch between i) the observations ($y$) and the corresponding model outputs, $H(x)$, (where $H$ is the model operator), and ii) the a priori ($x_b$) and optimised parameters ($x$), weighted by their error covariance matrices:

$$J(x) = (H(x) - y)^\mathrm{T} \mathbf{R}^{-1} (H(x) - y) + (x - x_b)^\mathrm{T} \mathbf{P}_b^{-1} (x - x_b). \tag{1}$$

$\mathbf{R}$ and $\mathbf{P}_b$ are the prior covariance matrices for the observation and parameter errors, respectively. The error covariance matrices are difficult to assess and thus neglected here, as in most studies (i.e. $\mathbf{R}$ and $\mathbf{P}_b$ are diagonal). The observation errors (variances) have been defined as the mean-squared difference between the observations and the prior model simulations. This reflects not only the measurement errors, but includes possibly significant model errors. The exact error values assumed for NEE and LE for each site can be found in Table S2.

As explained in the introduction, two classes of method to minimise the function $J(x)$ are compared; both are described below.

### 2.4.2 Gradient-based minimisation algorithm

For the gradient-based approach we chose the quasi-Newton algorithm L-BFGS-B to iteratively minimise the cost function (limited memory Broyden-Fletcher-Goldfarb-Shanno algorithm with bound constraints, see Byrd et al., 1995), referred as
BFGS. It was developed specifically for quasi nonlinear optimisation problems and allows direct assignment of bounds to each parameter.

Each iteration requires the evaluation of the cost function and its gradient with respect to each parameter. Generally, the gradient can be calculated with a finite-difference approximation (i.e. the ratio of the model outputs change to the value of the parameter perturbation that caused this change). However, it can be done more precisely using the tangent linear (TL)
model (linear derivative of the forward model). For ORCHIDEE the corresponding TL model has been derived with the automatic differentiation tool TAF (Transformation of Algorithms in Fortran; see Giering et al., 2005). It was used to



calculate the gradient for all parameters except for two phenology parameters, $K_{pheno,crit}$ and $C_{Tsen}$, associated to threshold functions. For these two parameters, the finite difference method is more appropriate and was used here.

The search is terminated when the relative change in the cost function becomes smaller than $10^{-4}$ during five consecutive iterations. On average the iterative process stopped within 30 iterations. Given that the computation of a TL model run is

about twice as long as the standard run of ORCHIDEE, the total computation time for one BFGS optimisation for one site with 30 parameters is equivalent to around 1800 standard model runs.

Gradient-based algorithms have a potential drawback with non-linear models as they are more likely to converge to a local minima of the cost function instead of the global one. In order to assess the extent of this problem we performed a set of independent assimilations starting with 16 different random initial first-guesses for the parameter vector, as in Santaren et al.

(2014) (see section 2.5).

### 2.4.3 Random search minimisation algorithm

For the random search approach, we chose the genetic algorithm (GA). The GA is a meta-heuristic optimisation algorithm, belonging to a larger class of evolutionary algorithms that follows the principles of genetics and natural selection (Goldberg, 1989; Haupt and Haupt, 2004). It considers the vector of parameters as a chromosome with each gene corresponding to a

given parameter. The algorithm works iteratively, filling a pool of $n$ chromosomes at each iteration. The starting pool is created with randomly perturbed parameters. The chromosomes at each following iteration are created from the randomly chosen chromosomes of the previous iteration by one of the two processes:

- exchange of the gene sequences of two parental chromosomes ("crossover" process),
- random perturbation of the selected genes of one parental chromosome ("mutation" process).

The resulting pool is then filled with $n$ best chromosomes from both parental and offspring pools, corresponding to the $n$ lowest cost function values. Furthermore at each step the chromosomes are ranked in the ascending order of the corresponding cost function values and random selection of the new parents is weighted by those ranks in order to guarantee that the best chromosomes produce more descendants ("selection" principle).

The genetic algorithm performance is sensitive to its particular configuration. In this study, we use the same GA

configuration as in Santaren et al. (2014) who tested different GA configurations and chose one, based on computational efficiency (to locate the cost function minimum) :

- number of chromosomes in the pool – 30;
- number of iterations – 40;
- crossover/mutation ratio – 4 : 1;

- number of gene blocks exchanged during crossover – 2;
- number of genes perturbed during mutation – 1.



For a better convergence of the cost function we reduce the ranges of parameter variations after the 30th iteration by a factor of 0.25 with the centre point at the current best guess value.

The GA does not require any gradient calculations; therefore, one chromosome rank estimate requires one standard model run and the total computational time for the GA assimilation for one site equals to 1200 runs.

### 2.4.4 Posterior uncertainty

Assuming Gaussian prior errors and linearity of the model in the vicinity of the solution, the posterior error covariance matrix of the parameters, $\mathbf{A}$, can be approximated by:

$$\mathbf{A} = [\mathbf{H}^T\mathbf{R}^{-1}\mathbf{H} + \mathbf{P}_b^{-1}]^{-1}, \tag{2}$$

where $\mathbf{H}$ is the model sensitivity (Jacobian) at the minimum of $J(x)$ (see for instance Tarantola, 1987). From $\mathbf{A}$ we can compute error correlations between parameters, a diagnostic that is used to evaluate the information content brought by the observations to discriminate each parameter.

### 2.5 Numerical experiments and performance metrics

For each data assimilation experiment we performed 16 independent runs, one of which has been performed with the standard model parameters as a first-guess and the other 15 with different first-guesses randomly selected within the range of variation of the parameters. We choose 16 sets based on the available computing resources but we verified for one case that increasing this number does not change the overall message. The set of 15 random first-guesses was kept identical for all experiments in order to guarantee the same first-guess values. However, for the GA such procedure is not essential, as the GA relies on a random generation process at each iteration. Both minimisation methods have been tested for two cases: an independent optimisation of the parameters at each site ("single-site" approach, SS) and a simultaneous optimisation of one common set of parameters at all sites belonging to the same PFT ("multi-site" approach, MS).

Overall, we defined the following set of numerical experiments:

- 16 single-site assimilation runs for each of the 78 sites that encompass all 7 PFTs with the gradient-based algorithm (BFGS) (noted below as $S_{BFGS}$);
- 16 single-site assimilation runs for each of the 78 sites that encompass all 7 PFTs with the genetic algorithm ($S_{Genetic}$);
- 16 multi-site assimilation runs for each of the 7 PFTs with the BFGS algorithm ($M_{BFGS}$);
- 16 multi-site assimilation runs for each of the 7 PFTs with the genetic algorithm ($M_{Genetic}$).

We address the impact of different first guesses for both the SS and MS optimisations for both methods in each section of the results. Additionally, for one PFT (Temperate Deciduous Broadleaf Forest – TempDBF) we performed the same four numerical experiments using pseudo-observations generated from outputs of the model with random parameters, in order to verify if the assimilation methods were finding the correct minima of the cost function. Such pseudo data are not biased by





observation uncertainties (no added uncertainties) or model discrepancies, and thus allow us to directly assess the performance and convergence properties of the optimisation schemes. Note that for these pseudo-data experiments the second term of the cost function (Eq. 1) was excluded.

The performances of the optimisation methods are compared using the reduction of the model–data root mean square
difference (RMSD) between the prior and the posterior, expressed as: $(1 - \mathrm{RMSD_{post}}/\mathrm{RMSD_{prior}}) \cdot 100\%$, as well as the range of posterior parameter values and their difference with the true value as estimated from the pseudo-observation tests.

## 3 Results and discussion

### 3.1 Model fit to the data

In order to investigate the differences between the minimisation schemes, Figure 1 presents a comparison of the overall
optimisation performances, as a summary diagnostic. It displays the prior and median posterior model–data RMSD for all sites used in the study (i.e., the median across the 16 first-guess tests). Additionally, the reader is referred to figure S1 in the Supplement for the results obtained at each site for each first guess with the prior and posterior model–data RMSD. We first note that both gradient-based and genetic algorithms successfully optimised the simulated NEE flux (for LE flux, see figure S1), reducing significantly the model–data misfit, in line with the results obtained by Kuppel et al. (2014) with the same
modelling framework.

### 3.1.1 Single site optimisation: comparative performance of the methods ($S_{BFGS}$ vs $S_{Genetic}$)

The first objective of the study is to assess which method performs best for a single site optimisation. Figure 2 (top panels) compares the performance of the two methods for all sites in terms of model–data RMSD reduction for the NEE fluxes. The RMSD reductions are expressed in percentage; they correspond to the median value (left plot) and the $5^{th}$-$95^{th}$ percentile
range ($R_{90}$) (right plot) across the 16 random first guess tests.

At all sites of all PFTs the single-site genetic algorithm ($S_{Genetic}$) provides better fit to the data (median across all 16 first guess tests) than the single-site gradient-based algorithm ($S_{BFGS}$), with only a few exceptions (see Figure S1 and Figure 2, upper left panel). The $S_{Genetic}$ leads up to 40% of additional model–data RMSD reduction compared to the $S_{BFGS}$ algorithm, with a mean improvement across all sites of 10%. In most cases the worst posterior RMSD value achieved with the $S_{Genetic}$
algorithm is better than the median value for the $S_{BFGS}$ method. The $S_{BFGS}$ approach is slightly better only at 8 sites with no more than 4% of additional RMSD reduction.

The spread of the results obtained with the use of different first guess parameters (16 tests) is also much lower for the $S_{Genetic}$ compared to the $S_{BFGS}$ method, for nearly all sites (Figure 2, upper right panel). On average the $R_{90}$ values are 3.6 times lower with the genetic algorithm – the $S_{Genetic}$ method is only slightly disadvantageous for 3 sites out of 78, although the
spread across the 16 first guesses remains low (<10%). Overall, the mean spread of the RMSD reduction for the $S_{Genetic}$ case





is around 10%. This clearly indicates that the genetic algorithm, following the set up described in section 2.4.3, is able to find nearly the same minimum of the cost function independently from the choice of the first guess parameters (with a significant improvement of the model-data misfit, see Figure 1). On the contrary, the results of the $S_{BFGS}$ method are highly dependent on the first guess model parameters, with a spread above 22% for half of the sites. Finally, if we compare the best

achieved RMSD reduction within the 16 first guess tests (see Tables S3 and S4 in the Supplement), we obtain similar performances for most of the sites of 5 PFTs out of 7, excluding TropEBF and BorDBF, where the $S_{Genetic}$ method still produces better results than the $S_{BFGS}$ method.

With the gradient-based method ($S_{BFGS}$), using the standard ORCHIDEE model parameters as a first guess does not guarantee an optimal reduction of the cost function – the corresponding posterior RMSD could be either the lowest or the

highest one of the 16 tests (these cases are shown by a circle, as opposed to a cross, in Figure S1 that displays the posterior RMSD for all sites and all 16 tests). Although we have used the same random parameter sets for each site of a given PFT, the "optimal" first-guess parameter set (i.e., the one providing the largest cost function reduction) differs between sites. It indicates that the shape of the cost function varies between sites and that there is no optimal prior first guess with the $S_{BFGS}$ gradient method, which is prone to getting caught in local-minima if the assumption of model quasi-linearity is not met.

Overall, this highlights one weakness of gradient-based methods and the need to perform several independent assimilations starting from different first-guess parameter values (see section 3.1.4).

### 3.1.2 Multi-site optimisation: comparative performances of the methods ($M_{BFGS}$ vs $M_{Genetic}$)

The reduction in NEE RMSD for the multiple site optimisations ($M_{BFGS}$ for the gradient-based method and $M_{Genetic}$ for the Genetic algorithm) is illustrated in the lower panels of Figure 2. First, the multi-site optimisations provide lower RMSD

reduction than the single-site optimisations (lower vs upper panels of Figure 2). This is the trade-off between fitting a specific site versus fitting an ensemble of sites representing more accurately the diversity of plant, soil and climate for a given PFT. We then notice that for few sites the RMSD is increasing after the optimisation (i.e. negative value of the RMSD reduction): 1 site for TropEBF and C3grass PFTs and 3 sites for TempENF and BorENF. On average these sites have only one or two years of data with large prior observation errors and thus a smaller weight in the cost function compared to the

other sites of the same PFT. This could thus explain that the optimisation degrade the model-data fit at these sites but it could also indicate that there is no optimal parameter set improving the model-data fit at all sites, suggesting the need to refine the PFT description and/or to improve the model structure.

The performance of the two methods differs between the PFTs. For TropEBF, TempENF and C3grass PFTs the genetic algorithm still provides on average significantly larger RMSD reduction than the gradient method, with a ratios between the

two of 2.0, 1.8 and 1.4, respectively. For the other four PFTs both methods show much smaller differences and for TempDBF the average RMSD reduction is exactly the same. Overall, at 50 sites (from 78) the $M_{BFGS}$ method provides performances comparable to the $M_{Genetic}$ method (RMSD reduction differ by less than 25%) or even slightly better (see





Tables S3/S4 in the Supplement for all numbers) and the mean additional RMSD reduction for the $M_{Genetic}$ method is only 6%.

The spread across random first guesses between the two methods is much more comparable than for the single-site optimisations. The $R_{90}$ values for the gradient-based method are only considerably larger for half of the sites than the genetic
algorithm method (with values up to 2 times larger), except for two PFTs (TempENF and TempDBF) where they are similar on average (Figure 2, lower right panel). This suggests that increasing the number of observations and/or capturing a greater range of sensitivity to the parameters acts to linearise or smooth the cost function, thereby ensuring that the gradient-based method does not get as caught in local minima as in the SS optimisations.

The last point to notice is that using the standard ORCHIDEE parameter set as a first guess with the $M_{BFGS}$ method always
leads to significant improvement of the model-data fit (see Figure S1) with RMSD reduction at the same level than the median RMSD reduction for the $M_{Genetic}$ method. This was not the case for the single-site optimisation (see Section 3.1.1). With a multiple-site optimisation the gradient-based scheme is less dependent on the first-guess, likely due to the smoothing of the cost function as discussed above and therefore performing several random first-guess tests is less needed. Overall for the multiple sites optimisations the choice of the minimisation algorithm seems thus less crucial than for the single site cases.

### 3.1.3 Benefit of multiple first guesses for a gradient-based approach

We now investigate more precisely the benefit of using several first guess tests. For a highly non-linear model the shape of $J$ may be complex and the gradient-based algorithm can easily get stuck in a local minima (as mentioned above). Therefore it is useful or necessary to perform a set of independent assimilation runs starting with different first-guesses. On the opposite side, a global search method is much less sensitive to the first-guess parameter values; specific choices related to the random
exploration of the parameter space by the algorithm (i.e. the mutation rate, the number of chromosome and the number of iteration in the case of the Genetic Algorithm; see section 2.4.3) become crucial and the use of different first-guesses is only an additional degree of freedom to explore the parameter space.

In order to analyse the influence of the number of first guess tests on the optimisation performance, for each configuration we bin the 16 first-guess optimisations in all possible groups of $n$ elements and defined for each group the maximum RMSD
reduction. We then take the mean values of all these maximums. The results are shown on Figure 3 for all configurations ($S_{BFGS}$, $S_{Genetic}$, $M_{BFGS}$ and $M_{Genetic}$) as a function of the number of first guesses ($n$). Only the mean across all PFTs is displayed and analysed. This illustrates the gain of adding more first-guess tests in the different configurations.

On average for the single-site configurations, the genetic algorithm ($S_{Genetic}$) with any number of first-guesses leads to 52±2% of RMSD reduction for the NEE fluxes; the increase in RMSD reduction between 1 and 16 first-guesses is small ($\approx 4\%$).
However, the gradient-based method ($S_{BFGS}$) needs 15 independent first-guess optimisations in order to reach the same level of RMSD reduction than with one first-guess of $S_{Genetic}$ and there is a substantial increase in the RMSD reduction from one to five first-guesses ($\approx 9\%$), while the improvement afterward is much smaller. For the SG method, the increase of the RMSD reduction is only significant from one to two first-guess tests.



For the multiple-site configurations, the performances of the gradient-based method are more similar to that of the genetic algorithm (although both with smaller RMSD reduction, as discussed in previous section). On average, the $M_{BFGS}$ method needs three first-guess tests to reach the performance of the $M_{Genetic}$ method obtained with one test. Like with the single-site configuration the increase of the RMSD reduction is substantial for the $M_{BFGS}$ method between 1 and 5 first-guesses ($\approx 10\%$) and much smaller afterwards.

Overall this analysis reveals that using several first-guesses is crucial for the gradient-based method and that at least 3 to 5 tests are required to obtain a RMSD within 5% of the optimal value obtained with 16 tests (for both $S_{BFGS}$ and $M_{BFGS}$). The genetic algorithm is much less sensitive to the number of first-guess tests performed. Using a multiple site configuration reduces the difference between the two algorithms and requires less random first-guess tests for the $M_{BFGS}$ method to achieve performances comparable to the $M_{Genetic}$ method. Note finally that, increasing the number of first guess tests to much larger values would lead to a convergence of the method performances.

## 3.2 Estimated parameter values

We now investigate how the choice of the minimisation method impacts the retrieved parameters. We only discuss the results of one PFT, temperate deciduous broadleaf forest (TempDBF), for a more in-depth investigation. We obtained similar findings for the other PFTs (see figure S4 in the Supplement). We selected TempDBF given that the RMSD reduction for this PFT is significant and that it includes a large number of sites (11). Before investigating the standard optimisations with real data, we discuss the results of the pseudo-observation experiments (see section 2.5) in order to analyse the ability of the different methods to retrieve "true" parameter values.

### 3.2.1 Estimating the correct posterior parameter value – pseudo-observation experiments

Figure 4 shows the estimated parameter values, the standard prior values and the "true" values for the pseudo-observation experiments with TempDBF. We display the mean and standard deviation across the 16 first-guess tests and for the single-site assimilations we take the mean across all sites.

On average both gradient-based and genetic algorithms are not able to retrieve exactly the true value for all parameters (i.e. values used to generate the pseudo data), although many of the parameters are relatively well estimated by the optimisation schemes. Out of 28 parameters, on average across all methods and all first-guesses, 14 have posterior values that are within 5% of the true value, using the physical parameter range to define the percentage (i.e., true value +/- 5% of the allowed range of variation), 6 parameters are between 5 and 10%, 7 between 10 and 20%, and one parameter ($Z_{crit,litter}$) is 29% far from the true value. The most constrained parameters (5% range with the small deviation around true value) are $G_{s,slope}$, $c_{Topt}$, $K_{pheno,crit}$, $C_{Tsen}$, $L_{age,crit}$, $\tau_{leafinit}$, $K_{soilC}$ and $Q_{10}$. We can thus expect more robust results with these parameters using real data. However, the uncertainties associated to the real data, and the fact that these uncertainties may not be adequately described in the observation error covariance matrix, will complicate the parameter retrieval (see MacBean et al. 2016). Note that for many





other parameters the error bars (standard deviation across all first-guess tests) encompass the true values ($V_{cmax}$, $c_{Tmin}$, $c_{Tmax}$, SLA, $K_{GR}$, $MR_{offset}$, $MR_{slope}$, $HR_{Hc}$, $HR_{Hmin}$, $Z_{decomp}$, $K_{albedo,veg}$, $K_{rsoil}$).

As it was already outlined in Santaren et al. (2014), the differences obtained between the true and optimised parameter values are likely due to equifinality problems (i.e., multiple solutions achieving the same overall global minimum of the cost

function) or the existence of local minimum where the algorithm is trapped. Given existing correlations and anti-correlations between the impacts of different parameters, it is not possible to retrieve all of them with the chosen set of observations (daily means of NEE and LE) and the optimisation frameworks that are used.

Differences between the methods exist but are not systematic. There are a few parameters that perform well only with one of the methods, both for single and multiple sites optimisations. For example, $c_{Topt}$, $\tau_{leafinit}$ and $Z_{decomp}$ are correctly estimated

with only the gradient-based approach. On the contrary, the generic algorithm perform much better for parameters like $LAI_{max}$, SLA, $L_{age,crit}$, $K_{LAIhappy}$, $K_{rsoil}$. A few parameters are not well estimated by both methods: the posterior values for $F_{stressh}$ and $HR_{Hb}$ differ from the true value by 20% of the physical range and $Z_{crit,litter}$ by 30%. The estimated errors (Eq. 2) associated to these poorly retrieved parameters are relatively large and for $F_{stressh}$ and $HR_{Hb}$ they are highly correlated with other parameters; we should thus be very cautious when interpreting their value with real data optimisations. Overall, we

observe that 19 parameters out of 28 are better estimated on average by the genetic algorithm ($S_{Genetic}$, $M_{Genetic}$) than by the gradient-based method ($S_{BFGS}$, $M_{BFGS}$). This is coherent with the results on the fit to the data (section 3.1), indicating that the gradient method is more likely stuck in local minima than the genetic algorithm.

### 3.2.2 Spread in parameter values across methods and first guess tests – real data experiments

Using real data, figure 5 displays the mean posterior estimates of the TempDBF parameters across all first-guess tests for the

four optimisation methods ($S_{BFGS}$, $S_{Genetic}$, $M_{BFGS}$ and $M_{Genetic}$). For most of the parameters the mean optimised values are relatively similar between the gradient-based and the genetic algorithms, with some exceptions. 12 parameters ($G_{s,slope}$, $c_{Topt}$, $c_{Tmin}$, $K_{pheno,crit}$, $C_{Tsen}$, $LAI_{max}$, $K_{LAIhappy}$, $Q_{10}$, $MR_{offset}$, $K_{GR}$, $HR_{Hc}$, $K_{z0}$) out of 28 show posterior differences between $S_{Genetic}$ and $S_{BFGS}$ methods that are lower than 5% of the physical range for each parameter, while 7 ($c_{Tmax}$, $F_{stressh}$, $K_{wroot}$, SLA, $\tau_{leafinit}$, $HR_{Hmin}$, $K_{albedo,veg}$) show differences between 10 and 20%, whereas $K_{rsoil}$ goes up to 36%. For $M_{Genetic}$ and $M_{BFGS}$ methods,

only 7 parameters are within the 5% variation range ($G_{s,slope}$, $C_{Tsen}$, $LAI_{max}$, $K_{z0}$ as for the single-site approaches, plus $HR_{Ha}$, $Z_{decomp}$, $Z_{crit,litter}$), and 14 parameters have differences over 10% (same list as for the single-site methods excluding $c_{Tmax}$, plus $c_{Topt}$, $c_{Tmin}$, $K_{pheno,crit}$, $L_{age,crit}$, $K_{LAIhappy}$, $K_{soilC}$ and $MR_{slope}$). Note that most parameters that are well constrained (i.e. small spread across the optimisation methods) correspond also to the most constraint one in the pseudo-observation experiments (like $G_{s,slope}$, $c_{Topt}$, $K_{pheno,crit}$, $C_{Tsen}$, $Q_{10}$, $HR_{Hc}$). However, few parameters demonstrate different behaviour between the real

and pseudo data tests (like $K_{LAIhappy}$, $MR_{offset}$, $LAI_{max}$).

A second feature is that for the single-site and the multiple-site cases, both algorithms (BFGS and GA) lead for most parameters to a similar dispersion of the posterior estimates from the ensemble of 16 first-guesses (see figure 5). However, for some parameters ($F_{stressh}$, SLA, $MR_{offset}$, $MR_{slope}$, $HR_{Ha}$, $HR_{Hb}$, $HR_{Hc}$, $HR_{Hmin}$, $Z_{crit,litter}$, $K_{albedo,veg}$, $K_{rsoil}$) the GA method



gives, surprisingly, much higher distribution of the posterior values. It corresponds to the parameters that have also not been correctly estimated in the pseudo-observation tests (see section above) with estimated value outside the 10% range around the true value. Although not intuitive, the random sampling over the parameter space with the genetic algorithm could explain that for each first guess the GA method may end up exploring a larger domain of the parameter space and thus

converge to more different parameter sets than the gradient-based method.

For each parameter, the dispersion obtained with the different first-guesses is lower for the multiple-site case compared to the single-site case, for both algorithms. On average the mean dispersion for the single-site approaches is 25% of the parameter range (with minimum and maximum spreads of 7.3% and 42%, respectively), and for the multiple-site approaches it is only 14% (between 3.4% and 32%). This reflects that with the multiple-site approach the shape of the cost function is

more smooth due to the combination of different NEE/LE responses to the parameter variations (differences induce mainly by different climate, soil type and plant species) which in turns facilitate the convergence to the global minimum. The only exception is for $K_{soilC}$, a parameter that is specific to each site even in the multiple-site approach (see section 2.2).

## 4 Summary and conclusion

Through this study, we compared the performances of two algorithms for the optimisation of the ORCHIDEE land surface

model parameters. The two algorithms belong to two different classes of method – L-BFGS algorithm for the gradient-based methods (BFGS) and the Genetic Algorithm (GA) for global random search methods. The two methods were used to optimise 28 parameters (16 being PFT dependant), using daily NEE and LE fluxes from 78 eddy covariance flux measurement sites. Both methods were run either independently at each site (single-site approach) or simultaneously at all sites of each PFT (multiple-site approach), running for each configuration 16 different tests where the prior parameter values

are selected randomly. The main findings are:

- For the single site optimisations (SS), the random search algorithm (GA) results in lower values of the cost function than the gradient-based method (BFGS) for nearly all sites.

- The difference between the GA and BFGS methods are smaller for multiple-site optimisations (MS) than at single sites (SS), due to a smoothing of the cost function shape with a greater number of observations.

- For the single-site tests, the spread in cost function minimum when performing the same tests with multiple first guesses in parameter space (16 random first guesses) is much larger for the BFGS than the GA methods, due to the higher likelihood that gradient based methods get stuck in local minima.

- For the multiple-site tests, the spread in the cost function from 16 first guesses are closer, although still higher for the BFGS than the GA methods.

- With the BFGS method, performing the same tests with at least 5 different first guess parameters ensure a reduction of the cost function comparable to the one obtained with random search GA method.




- The GA results in a closer approximation of the true posterior parameter value than BFGS with the pseudo-observation tests, for both the SS and MS experiments.

The computing cost of the BFGS and GA algorithms were on average relatively similar, when considering all experiments, although slightly higher for the BFGS algorithm. With BFGS, the cost depends at which iteration the convergence criterion

is met (see section 2.4.2). For the random search GA method, the value of the cost function may be further decrease by increasing the number of iterations (currently at 40). We choose a set up for the random search method that led to similar computing cost than the gradient-based method, but this could be revised depending on the cost of a single-site model simulation (currently around 20-30 seconds with ORCHIDEE for one year using one standard processor depending on the number of output variables).

Most of the differences between the BFGS and GA algorithms are related to the shape of the cost function, in part controlled by the non-linearity of the model. Our results can thus be extrapolated to other land surface models, provided that they have similar complexity and level of non-linearity. With single-site optimisations, we advise to use a random search method, the Genetic Algorithm being just one possibility. If a gradient-based method was preferred, we strongly recommend performing at least several tests (5 or more) with different random first guess parameter values. With multiple-site optimisations, the use

of a gradient-based method is less critical, due to a smoother cost function with the addition of data. In this case, pseudo data experiments, with known true parameter values, help assessing the strength of the method with the selected model and observations; they reveal how many and which parameters may likely be poorly constrained because of the existence of local minima or a broad flat minimum of the cost function. Finally, note that with a random search method, we can use more easily any assumption for the probability distribution function of the prior observation/parameter uncertainties.

The results obtained in this study with eddy covariance flux measurements are likely to hold with other, site-specific, in situ or remote sensing observations (i.e., satellite vegetation activity data, carbon pool measurements, etc.), although similar investigations are needed to quantify precisely the differences between methods. On the other hand, using global simulations (i.e. all grid cells) is likely to produce smoother cost functions and thus to favour gradient-based methods, at least from a computational point of view (especially if an adjoint model is available to derive efficiently the gradient of the cost function

with respect to all parameters). This is illustrated in Ziehn et al. (2012) with the assimilation of atmospheric $CO_2$ data in BETHY LSM. Additional studies to characterise the level of smoothness of the cost function as a function of model complexity, data types, approach (from single-site to global optimisation) would nevertheless be beneficial and provide other practical guidelines to newcomers in their choice of parameter optimisation method.

**Code availability**

The ORCHIDEE model code and the run environment are open source and available at http://orchidee.ipsl.fr. The tangent linear version of the ORCHIDEE model has been generated using commercial software (TAF; http://www.fastopt.com/products/taf), thus only the "forward" version of the ORCHIDEE model is available for sharing. The



optimisation tool is available through a dedicated web site for data assimilation with ORCHIDEE (http://orchidas.lsce.ipsl.fr). Nevertheless readers interested in running ORCHIDEE and/or optimisation tool are encouraged to contact the corresponding author for full details and latest bug fixes.

**Acknowledgements**

This work used eddy covariance data acquired by the FLUXNET community and in particular by the following networks: AmeriFlux (U.S. Department of Energy, Biological and Environmental Research, Terrestrial Carbon Program (DE-FG02-04ER63917 and DE-FG02-04ER63911)), AfriFlux, AsiaFlux, CarboAfrica, CarboEuropeIP, CarboItaly, CarboMont, ChinaFlux, Fluxnet-Canada (supported by CFCAS, NSERC, BIOCAP, Environment Canada, and NRCan), GreenGrass, KoFlux, LBA, NECC, OzFlux, TCOS-Siberia, USCCC. We acknowledge the financial support to the eddy covariance data

harmonisation provided by CarboEuropeIP, FAO-GTOS-TCO, iLEAPS, Max Planck Institute for Biogeochemistry, National Science Foundation, University of Tuscia, Université Laval, Environment Canada and US Department of Energy and the database development and technical support from Berkeley Water Center, Lawrence Berkeley National Laboratory, Microsoft Research eScience, Oak Ridge National Laboratory, University of California – Berkeley and the University of Virginia.

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



**Table 1: List of optimised parameters with description and default model values.**

| Parameter | Description | Plant functional type* | | | | | | |
|---|---|---|---|---|---|---|---|---|
| | | Trop EBF | Temp ENF | Temp EBF | Temp DBF | Bor ENF | Bor DBF | C3 grass |
| **Photosynthesis** | | | | | | | | |
| $V_{cmax}$ | Maximum carboxylation rate ($\mu$mol m$^{-2}$ s$^{-1}$) | 65 | 35 | 45 | 55 | 35 | 45 | 70 |
| $G_{s,slope}$ | Slope of the Ball-Berry relationship | 9 | 9 | 9 | 9 | 9 | 9 | 9 |
| $c_{Topt}$ | Optimal photosynthesis temperature (ºC) | 37 | 25 | 32 | 26 | 25 | 25 | 27.25 |
| $c_{Tmin}$ | Minimal photosynthesis temperature (ºC) | 2 | –4 | –3 | –2 | –4 | –4 | –3.25 |
| $c_{Tmax}$ | Maximal photosynthesis temperature (ºC) | 55 | 38 | 48 | 38 | 38 | 38 | 41.13 |
| **Soil water availability** | | | | | | | | |
| $F_{stressh}$ | Parameter reducing the hydric limitation of photosynthesis | 6 | 6 | 6 | 6 | 6 | 6 | 6 |
| $K_{wroot}$ | Parameter describing exponential root profile (m$^{-1}$) | 0.8 | 1 | 0.8 | 0.8 | 1 | 1 | 4 |
| **Phenology** | | | | | | | | |
| $K_{pheno,crit}$ | Parameter controlling the start of growing season | — | — | — | 1 | — | 1 | 1 |
| $C_{Tsen}$ | Offset controlling the start of senescence (ºC) | — | — | — | 12 | — | 7 | — |
| $LAI_{max}$ | Maximum leaf area index (m$^2$ m$^{-2}$) | 7 | 5 | 5 | 5 | 4.5 | 4.5 | 4.5 |
| SLA | Specific leaf area (m$^2$ g$^{-1}$) | 0.0154 | 0.0093 | 0.02 | 0.026 | 0.0093 | 0.026 | 0.026 |
| $L_{age\_crit}$ | Mean critical leaf lifetime (days) | 730 | 910 | 730 | 180 | 910 | 180 | 120 |
| $K_{LAIhappy}$ | Minimum fraction of $LAI_{max}$ below which the carbohydrate reserve is used | 0.5 | 0.5 | 0.5 | 0.5 | 0.5 | 0.5 | 0.5 |
| $\tau_{leafinit}$ | Time to attain initial foliage (days) | 10 | 10 | 10 | 10 | 10 | 10 | 10 |
| **Respirations** | | | | | | | | |
| $K_{soilC}$ | Multiplicative factor for initial soil carbon stocks | 1 (site dependant) | | | | | | |
| $Q_{10}$ | Parameter determining the temperature dependency of the heterotrophic respiration | 1.99372 | | | | | | |
| $MR_{offset}$ | Offset and slope of the linear relationship between temperature and maintenance respiration | 1 | | | | | | |
| $MR_{slope}$ | | 0.12 | 0.16 | 0.16 | 0.16 | 0.16 | 0.16 | 0.16 |
| $K_{GR}$ | Fraction of biomass available for growth respiration | 0.28 | 0.28 | 0.28 | 0.28 | 0.28 | 0.28 | 0.28 |





| Respirations responses on water availability | | |
|---|---|---|
| $HR_{Ha}$ | Parameters of the quadratic function determining the moisture control of the heterotrophic respiration (HR) | −1.1 |
| $HR_{Hb}$ | | 2.4 |
| $HR_{Hc}$ | | −0.29 |
| $HR_{Hmin}$ | Minimum value of the HR function | 0.25 |
| $Z_{decomp}$ | Scaling depth determining the effect of soil water on litter decomposition (m) | 0.2 |
| $Z_{crit\_litter}$ | Scaling depth determining the litter humidity (m) | 0.08 |
| Energy balance | | |
| $K_{z0}$ | Reference rugosity length (m) | 0.0625 |
| $K_{albedo\_veg}$ | Multiplicative factor of vegetation albedo | 1 |
| $K_{rsoil}$ | Resistance to bare soil evaporation | 3.3 |

\* TropEBF — tropical evergreen broadleaf forest; TempENF — temperate evergreen needleleaf Forest; TempEBF — temperate evergreen broadleaf forest; TempDBF — temperate deciduous broadleaf forest; BorENF — boreal evergreen needleleaf forest; BorDBF — boreal deciduous broadleaf forest; C3 grass — C3 grassland.



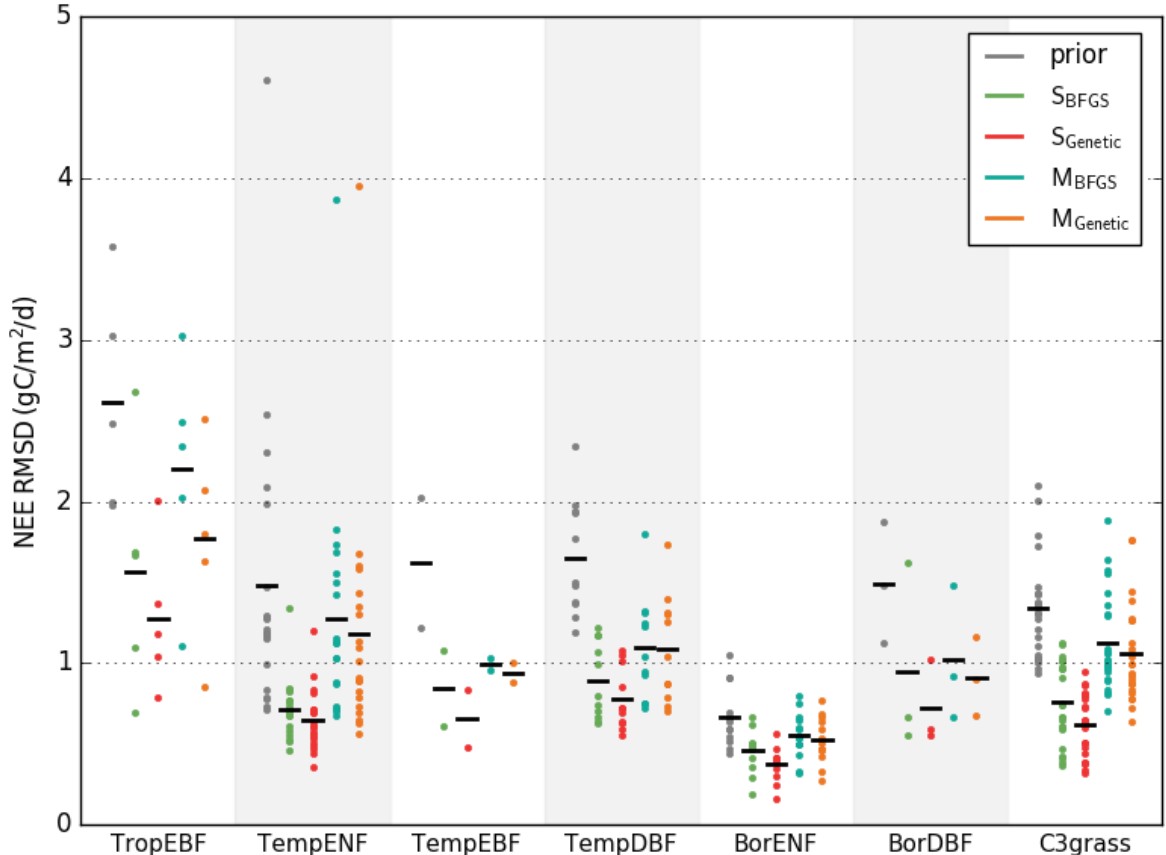

Figure 1: Model–data RMSD for the daily NEE time series at each site grouped by PFT. The prior RMSD values are shown in grey, followed by the median posterior RMSD values obtained within 16 optimisation tests with random first-guess parameter values for each optimisation method. Black horizontal bars show the mean value across the sites for each PFT.

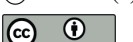



**Figure 2: Comparison of the performances between the gradient-based and genetic methods in terms of model–data RMSD reduction (%) obtained for the daily NEE fluxes. The left panels show the medians across the results obtained within 16 optimisation tests with random first-guess parameter values for each site; the right panels – the spreads between the 5th and 95th percentiles ($R_{90}$) of the same distributions.**





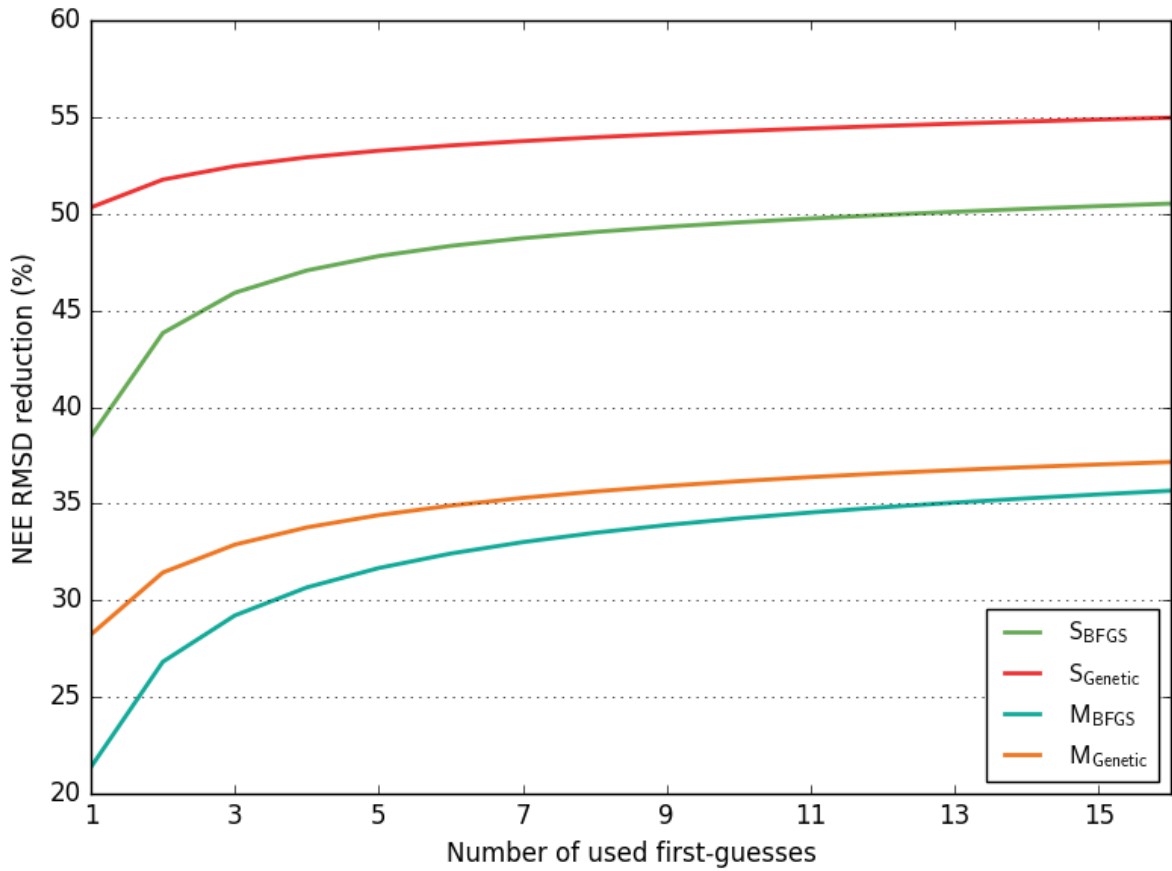

**Figure 3: Model–data RMSD reduction (in %) obtained for the NEE fluxes as a function of the number of runs performed with random first guess parameter values (for each configuration, $S_{BFGS}$, $S_{Genetic}$, $M_{BFGS}$ and $M_{Genetic}$). For each number of first guesses (X-axis) all possible combinations across the 16 optimisation tests (i.e. 16 first-guesses) are considered and the maximum RMSD reductions are calculated; the mean of these maximums is reported on the Y-axis. The results are averaged across all sites for each configuration.**



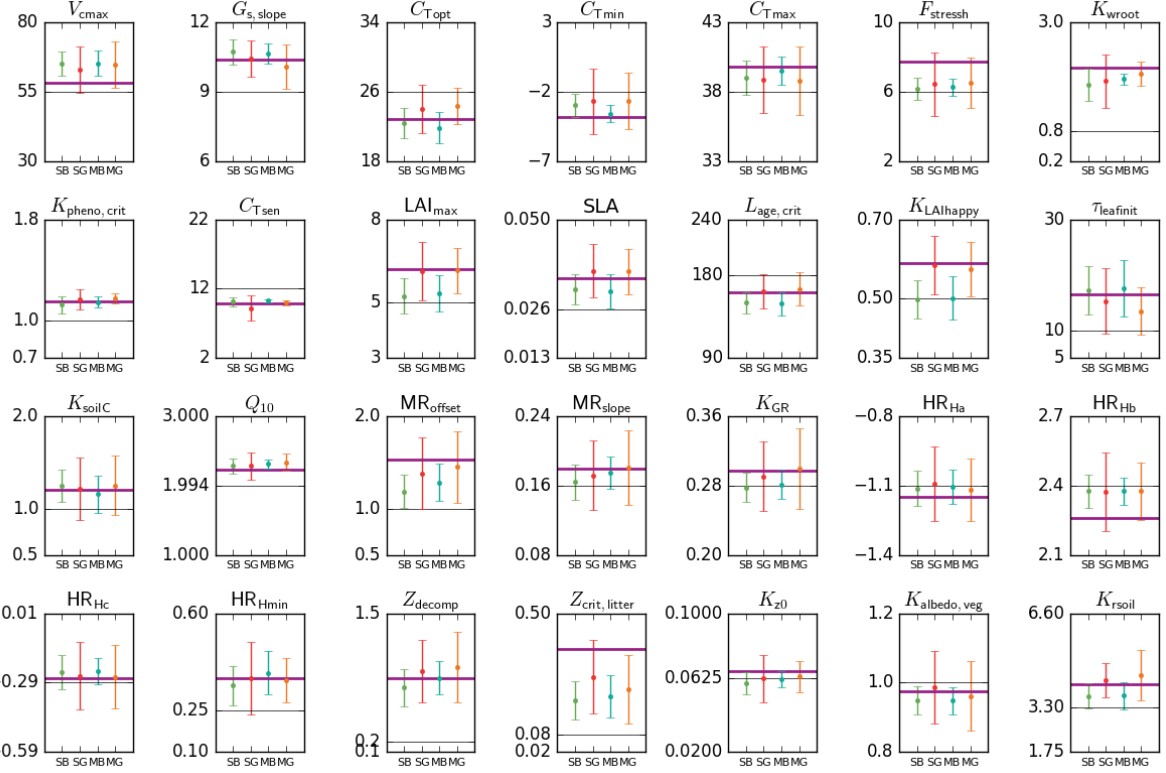

**Figure 4: Parameter estimates from a pseudo-data experiment for the temperate deciduous broadleaf forest PFT. Result for the four different methods: $S_{BFGS}$ (SB), $S_{Genetic}$ (SG), $M_{BFGS}$ (MB) and $M_{Genetic}$ (MG) are displayed: mean value and standard deviation across 16 random first guess tests. The thick horizontal purple lines correspond to the "true" parameter values (defined randomly); the thin grey horizontal lines correspond to the prior value, which is equal to the default ORCHIDEE model value. The vertical axis limits represent the range of parameter variation.**





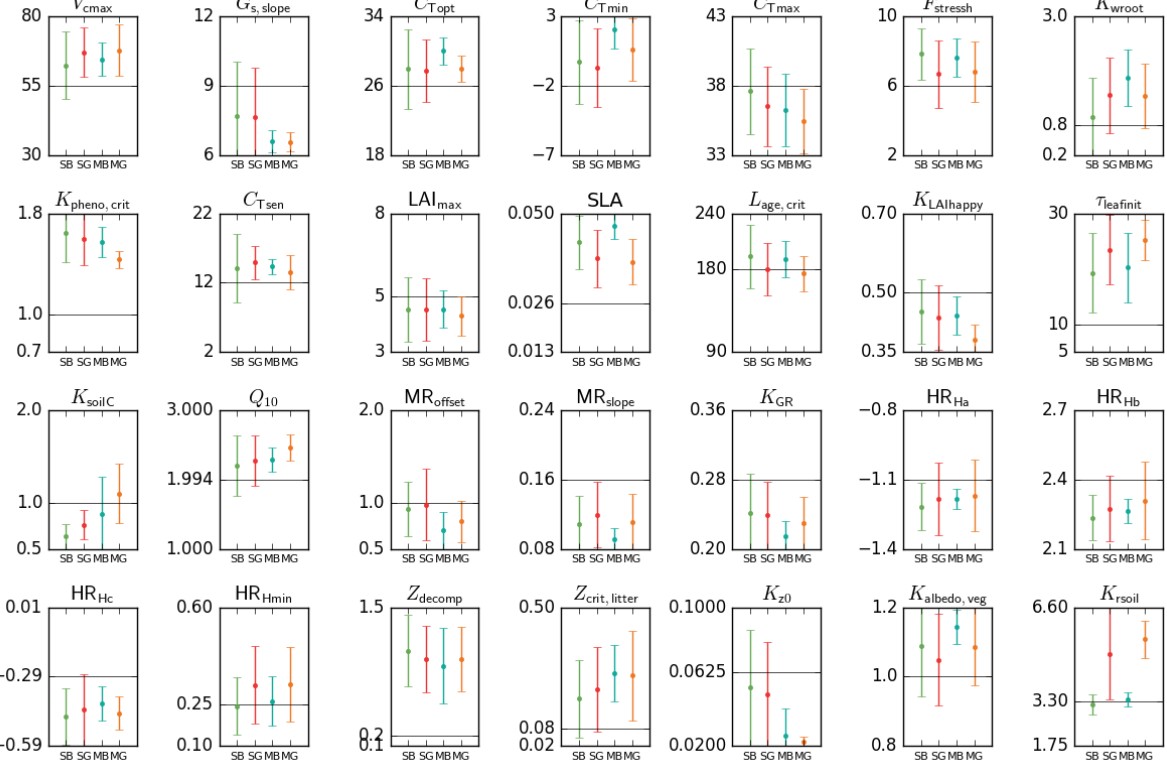

**Figure 5: Mean posterior model parameter values for temperate deciduous broadleaf forest PFT. Result for the four different methods: $S_{BFGS}$ (SB), $S_{Genetic}$ (SG), $M_{BFGS}$ (MB) and $M_{Genetic}$ (MG) are displayed: mean value and standard deviation across 16 random first-guess tests. Grey horizontal lines represent prior values, which are equal to the default ORCHIDEE model values. The vertical axis limits represent the range of parameter variation.**