# Peer review of "Land surface model parameter optimisation using in-situ flux data: comparison of gradient-based versus random search algorithms"

_Geoscientific Model Development, 2018_

## Short Comment (SC1) · 31 Jul 2018

Dear authors,

In my role as Executive editor of GMD, I would like to bring to your attention our Editorial version 1.1:

http://www.geosci-model-dev.net/8/3487/2015/gmd-8-3487-2015.html

This highlights some requirements of papers published in GMD, which is also available on the GMD website in the 'Manuscript Types' section:

http://www.geoscientific-model-development.net/submission/manuscript_types.html

[Figure]

In particular, please note that for your paper, the following requirements have not been met in the Discussions paper:

- "The main paper must give the model name and version number (or other unique identifier) in the title."

- "If the model development relates to a single model then the model name and the version number must be included in the title of the paper. If the main intention of an article is to make a general (i.e. model independent) statement about the usefulness of a new development, but the usefulness is shown with the help of one specific model, the model name and version number must be stated in the title. The title could have a form such as, "Title outlining amazing generic advance: a case study with Model XXX (version Y)"."

Therefore please add the ORCHDEE model (including version number) in the title of your publication, e.g. "Land surface model parameter optimisation using in-situ flux data: comparison of gradient-based versus random search algorithms: a case study using ORCHIDEE version x.y"

Yours,

Astrid Kerkweg

---

## Referee Comment (RC1) · Anonymous Referee #1 · 17 Aug 2018

This study digs deep into parametric uncertainty in the ORCHIDEE land surface model, using data assimilation to integrate FLUXNET observations. The authors provide a useful primer for future work on land surface model parameter optimization. This paper could use some reorganization and clarification, as well as adding additional points of discussion to some of the key results. I recommend publication after these and other comments are addressed.

General comments:

There are a couple of key results that could use more discussion (either in the appropriate place in Section 3 or at the end in Section 4). For example, why do certain

parameters have different responses to different optimization methods? What are the limitations of multiple-site optimizations? Adding some discussion could really solidify the take home points of this paper and provide relevance to other land surface models and parameter optimization studies.

Though the focus is parametric uncertainty, I think the paper would also benefit from a brief discussion of model structural uncertainty, with relevant details specific to OR-CHIDEE. This addition would provide useful context for discussing the results (e.g., Page 10, Line 25).

I think it would be useful to include some background on choice of model parameters and how their sensitivity was assessed, as this is a key step to narrowing the parameter space.

Minor note, but it would be preferable to have the line numbers continuously increasing throughout the document so identifying page numbers in the specific comments is not necessary.

Specific comments:

Page 3, Line 2: "probability distribution function"

Page 3, Line 6: What are the limitations of the Gaussian assumption? Could some parameters have different PDFs?

Page 3, Line 19: What defines "excessive" here? Can you give examples of the number of parameters explored in these studies, and how they compare to the dimensionality of your problem?

Page 3, Line 33: L-BFGS_B should be L-BFGS-B?

Page 4, Line 9: Word choice "exploited" could be changed to "utilized".

Page 4, Line 13: Change to "a few".

Page 4, Line 16: ORCHIDEE should be defined at first mention (Page 3, Line 23 and in abstract).

Page 4, Line 19: Why "possibly"? Has the use of ORCHIDEE on thousand-year timescales not been proven?

Page 5, Line 5ff: A few lines about the choice of parameter ranges and sensitivity assessment would be useful here (even if in the supplemental to go along with Table S1).

Page 5, Line 13: Repetitive here to again mention land use change in parentheses.

Page 5, Line 20: How useful is 1 year of station data? Only one observed seasonal cycle, especially relevant as optimization is on seasonal/annual time scales. Relevant also at Page 10, Line 23.

Page 5, Line 27: Please add a sentence explaining why you would expect the Bowen ratio to be constant.

Page 6, Line 19: Should be Tables S3-S4.

Page 6, Line 23: L-BFGS-B acronym should be defined at first mention, in introduction Page 3, Line 30 (and abstract).

Page 7, Line 1: Change "to threshold" to "with threshold".

Page 9, Line 13: Would be nice to include the equivalent of Figure 1 for LE flux. Figure S1 has it broken out by PFT but not a summary figure.

Page 10, Line 25: This is a key point – how do model structural uncertainties get in the way of multiple site optimizations? What are the limits to finding an optimal parameter set across multiple sites? (Also "degrade" should be "degrades"?)

Section 3.1.3: This section could be moved earlier in the paper as it is referenced in earlier parts of the results. Overall the flow of the results section could be improved.

[Figure]

Page 11, Line 32: First use of "SG" – replace with S_Genetic? That is the abbreviation used in other parts of the text.

Page 12, Line 27: Why does the pseudo-observation experiment perform poorly for Z_crit,litter? Why does it perform better for other parameters?

Page 13, Line 8ff: What drives different parameters to respond better or worse to different optimization methods?

Page 13, Line 12: The error in Z_crit,litter was mentioned on the previous page as 29%, please clarify.

Page 13, Lines 27, 33: Same question as previous section; what drives different responses in different parameters?

Page 14, Line 20ff: Some grammar issues in the bullet points, and throughout this section (e.g., Page 15, Lines 5, 16 and 23).

Page 15, Line 18: I think you should mention this point in Section 2.4.3.

Figure 1: In legend, use dots instead of lines to help guide the reader.

Figure 3: Add error bars here, the uncertainty on the RMSD reduction is referenced in the text (e.g., Page 11, Line 28).

Table S1: Missing units for parameters, as applicable.

Tables S3, S4: PFTs are numbered but not specified by name.

Figure S1: Have the legend in one panel and get rid of them elsewhere, they are just distracting/overlapping data points.

Figures S2, S3: Where in the main text are these figures referenced?

Figure S3: Add error bars, same comment as Figure 3.

Figure S4: If some parameters do not apply to certain PFTs (e.g., K_pheno,crit), why

are they optimized for that PFT? Is this an error in Table 1 and/or Table S1?

---

## Referee Comment (RC2) · Anonymous Referee #2 · 15 Sep 2018

General evaluation:

The manuscript is of relevance for GMD. The study is solid and I have only few points for additional clarification. The paper is also structured and written well. The amount of tables and figures is restricted to the absolutely necessary. A few more figures could have been included, but in general I like that the number of figures is not so large.

Nevertheless, the introduction could also elaborate on further parameter estimation techniques which also could show a better performance than the ones addressed here, in particular concerning uncertainty characterization (ensemble methods) and handling non-Gaussianity (e.g., particle filter methods or Markov Chain Monte Carlo methods,

for example the work by Post et al., 2017, JGR-Biogeosciences).

I recommend therefore minor revision of the manuscript.

Specific Comments:

P2, L12, L13, L15: These error sources (vegetation and soil spatial information versus parameter values in model) are partially the same thing? Please clarify.

P5, L7: Could you provide more details? How is the sensitivity study conducted? Reference?

P5, L22: Can you provide more details? How many sites were disregarded?

P6, L31: How automatic was the automatic differentiation? Can you provide more details on the additional coding and time which was required?

P8, Eq. 2: How critical is the Gaussian assumption? Non-Gaussianity of parameters can be expected.

P14, L30: How do you know whether this number (5) is not case dependent? In case of multiple sites, many tests could be carried out, as the parameters need to be determined just once (per PFT of course). Why would one need to impose restrictions, and would it not be better to use a larger number of initial guesses for cases with parameter estimation for multiple sites?

Editorial:

P1, L26: trapped instead of trap.

P10, L25: degrades instead of degrade

P10, L29: skip "a"

P10, L32: "reductions" instead of "reduction".

P11, L11: "at the same level as" instead of "at the same level than".

P11, L25: "maxima" instead of "maximums". The results are shown in?

P11, L31: "as" instead of "than".

P13, L5: "minima" instead of "minimum".

P13, L28: Change to: "the most constraint ones"

P14, L17: "dependent"

P14, L30: change to: "ensures".

P15, L5: change to "decreased".

---

## Author Comment (AC3) · 29 Oct 2018

Dear Referee,

We would like to thank you for the review and for your interest in this work. Below we provide the answers to your general and specific comments.

Referee: Nevertheless, the introduction could also elaborate on further parameter estimation techniques which also could show a better performance than the ones addressed here, in particular concerning uncertainty characterization (ensemble methods) and handling non-Gaussianity (e.g., particle filter methods or

**Markov Chain Monte Carlo methods for example the work by Post et al., 2017, JGR-Biogeosciences).**

Answer: Indeed it is possible that other methods may provide a better performance than the ones addressed here. However, these studies have not been performed, so we do not know which method performs the best. It would certainly be a good extension to this study to test other methods, and it would be good for the DA community at large to perform these kinds of method sensitivity tests across a range of model complexities. We have discussed the differences in methods extensively – including MCMC and particle filter methods – in the text (original submission P3 lines 7-125). However, we have added a reference to Post et al. (2017) earlier in the text when we talk about assumptions of Gaussian PDFs (P3 Line 4) and references to Post et al., (2017), Richardson et al. (2010), Pinnington et al. (2016) in this paragraph to highlight, as suggested by the reviewer, that there are other methods out there to consider. We further add these sentences before listing the key questions we are investigating:

"Note that this study does not aim to provide an exhaustive examination of all methods belonging to both classes of inversion algorithms (as previously described), nor do we presume to have chosen the best method belonging to each class. We simply choose to test two methods belonging to each class that have already been used to estimate parameters of the ORCHIDEE model. A further examination of the benefits of all methods would be beneficial to the LSM and DA community, but is outside the scope of this study."

Referee: P2, L12, L13, L15: These error sources (vegetation and soil spatial information versus parameter values in model) are partially the same thing? Please clarify.

Answer: We believe that these error sources relate to different aspects of the model. The vegetation and soil spatial information represents the so-called forcing data sets (like the meteorology); they correspond to global maps derived partly from satellite

**GMDD**
earth observation missions and contain the necessary information to cluster the different types of soils and vegetation all over the globe. The parameter values correspond to specific and chosen formulations of the different processes controlling the carbon, water and energy budgets in the model. They are thus internal to the model while the forcing data are external and may easily vary depending on the region where the model is applied. It is thus helpful to distinguish these different error sources and to classify them into separate groups.

We have thus only slightly changed the text to be more precise.

**Referee: P5, L7: Could you provide more details? How is the sensitivity study conducted? Reference?**

Answer: The choice of the model parameters was done based on sensitivity tests of the data used in optimization (net CO2 ecosystem exchange (NEE) and latent heat (LE) fluxes) with respect to all related ORCHIDEE parameters. This was done in previous works, as already cited in the paper and based on the so-called Morris method (Morris, 1991), which ranks the variability of "elementary effects" of the sampled parameters with respect to their impact on the model outputs. This information together with the new reference has been added to the reviewed text.

**Referee: P5, L22: Can you provide more details? How many sites were disregarded?**

Answer: Based on an original list of 252 sites from the La Thuile dataset (Baldocchi, 2001), we conducted a screening as described in the paper resulting in the selection of the 78 sites used in this study. Thus, we have disregarded around 70% of the sites. The total number of the sites in the original database has been added to the reviewed text. Additional reference to the article devoted to the PFT refinement subject is also added: "Note that Peaucelle et al. (in review) explored with the same model how to account for plant functional trait variability, refining the PFT distribution".
**Referee: P6, L31: How automatic was the automatic differentiation? Can you provide more details on the additional coding and time which was required?**

Answer: The differentiation has been generated by the TAF automatic differentiation tool from the FastOpt company (see http://www.fastopt.com/). However, the success of automatic differentiation largely depends on the cleanliness of the model code and to a certain extent on the structure of the code. Our group had spent a large effort on cleaning and making the initial ORCHIDEE model code, suitable for the TAF software. Note that a specific document with recommended coding guidelines has been built as the result of this work. Additionally, some input/output peace of code was also inserted to handle the tangent linear variables that are differentiated through TAF. Overall, this work required a strong investment of one software engineer and it took us around two years to have a working tangent linear model.

We thus added in the text one sentence to resume the committed effort: "For OR-CHIDEE the corresponding TL model has been derived with the automatic differentiation tool TAF (Transformation of Algorithms in Fortran; see Giering et al., 2005), following code cleaning and structural adjustments (without changing the physics) to allow the use of TAF (a significant effort that took around two years)"

**Referee: P8, Eq. 2: How critical is the Gaussian assumption? Non-Gaussianity of parameters can be expected.**

Answer: The currently implemented data assimilation technique relies on the assumption that the errors on both the parameters and the observations have Gaussian PDF. In this case, the resolution of the inverse problem, following a Bayesian framework, is equivalent to the minimization of a quadratic cost function. This Gaussian hypothesis significantly simplifies the interpretation of the minimum of the quadratic cost-function (i.e. being the mean of the posterior parameter PDF). If some parameters would have other PDF, the L-BFGS-B minimization procedure would not provide a meaningful value to describe the posterior PDF. However, for the GA this restriction does not hold, as Interactive comment

we could use the ensemble of model trajectories to describe the posterior PDF. Additionally, the Gaussian hypothesis is also central to calculate a posterior parameter uncertainty that fully describes (together with the mean value) the shape of the PDF, and this hypothesis also allows to compute the posterior uncertainty with a simple matrix formulation (see for instance Tarantola, 2005). Such hypothesis is used in many inversion problems.

However, we agree that non-Gaussianity may be the case for some parameters and that it could thus partially bias the overall parameter optimization with Gaussian assumption. It is nevertheless out of the scope of this paper to investigate non Gaussian errors. Moreover, in the case of ORCHIDEE we have shown in an earlier study (Santaren et al., 2007) that most parameter errors follow Gaussian distributions. We have inserted these points at the end of section 2.4.1:

"Note that using non-Gaussian errors would significantly complicate the application of one class (gradient-based) and is thus out of the scope of this study and that Santaren et al. (2007) have shown with a previous version of ORCHIDEE that most parameter errors follow Gaussian distributions."

Referee: P14, L30: How do you know whether this number (5) is not case dependent? In case of multiple sites, many tests could be carried out, as the parameters need to be determined just once (per PFT of course). Why would one need to impose restrictions, and would it not be better to use a larger number of initial guesses for cases with parameter estimation for multiple sites?

Answer: We agree with the reviewer that the larger the number of first-guess tests is, the more robust the results will be. However, this has to put in regards to the computing time that is increasing proportionally to the number of first-guesses. Note that the computing time does not vary substantially between the single and the multi-site cases as even for the multi-site case the model has to be run at all sites for each iteration. This chosen number (5) comes from a first order analysis of the results presented in

GMDD
Figure 3 with our particular model and set of observations. It should be seen as a first order and prior suggestion. For any other model and parameter inversion exercise it could of course be different. Note that this summary point is primarily to stress the fact that using only one first-guess inversion is very risky with a gradient-based method.

Overall, we agree that this statement needs to be put into a more general context and perspective. We have thus changed the text to include the above elements.

**Referee:**

- P1, L26: trapped instead of trap.
- P10, L25: degrades instead of degrade
- P10, L29: skip "a"
- P10, L32: "reductions" instead of "reduction".
- P11, L11: "at the same level as" instead of "at the same level than".
- P11, L25: "maxima" instead of "maximums". The results are shown in?
- P11, L31: "as" instead of "than".
- P13, L5: "minima" instead of "minimum".
- P13, L28: Change to: "the most constraint ones"
- P14, L17: "dependent"
- P14, L30: change to: "ensures".
- P15, L5: change to "decreased".

Answer: All mentioned typos are corrected in the text. We thank very much the referee for the thorough and attentive reading.

Best regards,

**GMDD**
Vladislav Bastrikov

**References**

Baldocchi, D., Falge, E., Gu, L., Olson, R., Hollinger, D., Running, S., Anthoni, P., Bernhofer, C., Davis, K., and Evans, R.: FLUXNET: a new tool to study the temporal and spatial variability of ecosystem-scale carbon dioxide, water vapor, and energy flux densities, B. Am. Meteorol. Soc., 82, 2415-2434, 2001. Morris, M.D.: Factorial sampling plans for preliminary computational experiments. Technometrics, 33(2), 161-174, 1991.

Giering, R., Kaminski, T., and Slawig, T.: Generating efficient derivative code with TAF, Fut. Gen. Comp. Syst., 21, 1345-1355, doi:10.1016/j.future.2004.11.003, 2005.

Morris, M.D.: Factorial sampling plans for preliminary computational experiments. Technometrics, 33(2), 161-174, 1991.

Peaucelle, M., Bacour, C., Ciais, P., Peylin, P., Vuichard, N., Kuppel, S., Peñuelas, J.: Exploring plant functional traits variability with a terrestrial biosphere model, in review.

Pinnington, E. M., Casella, E., Dance, S. L., Lawless, A. S., Morison, J. I. L., Nichols, N. K., Wilkinson, M. and Quaife, T. L.: Investigating the role of prior and observation error correlations in improving a model forecast of forest carbon balance using Four Dimensional Variational data assimilation. Agricultural and Forest Meteorology, 228-229, 299-314, doi:10.1016/j.agrformet.2016.07.006, 2016.

Post, H., Vrugt, J.A., Fox, A., Vereecken, H. and Hendricks Franssen, H.J.: Estimation of Community Land Model parameters for an improved assessment of net carbon fluxes at European sites. Journal of Geophysical Research: Biogeosciences, 122(3), 661-689, 2017.

Richardson, A.D., Williams, M., Hollinger, D.Y., Moore, D.J., Dail, D.B., Davidson, E.A., Scott, N.A., Evans, R.S., Hughes, H., Lee, J.T. and Rodrigues, C.: Estimating parameters of a forest ecosystem C model with measurements of stocks and fluxes as joint

GMDD
constraints. Oecologia, 164(1), 25-40, 2010.

Santaren, D., Peylin, P., Viovy, N., and Ciais, P.: Optimizing a process-based ecosystem model with eddy-covariance flux measurements: A pine forest in southern France, Global Biogeochem. Cycles, 21, GB2013, 2007.

Tarantola, A.: Inverse problem theory and methods for model parameter estimation, Siam, 2005.

**GMDD**

---

## Author Response (AR1)

**Response to Executive Editor**

Dear Astrid,

Thank you for your rightful remark. We have overlooked this point. The work was done with the ORCHIDEE model version 1.9.5.2. We now add the reference to the model itself and to the version used in the title of the paper. We also put the model version number in the main text.

Best regards, Vladislav Bastrikov

**Response to Anonymous Referee #1**

Dear Referee,

Thank you for your review and for your interest in this study. Below we provide the answers to your general and specific comments in sequential order.

Referee: There are a couple of key results that could use more discussion (either in the appropriate place in Section 3 or at the end in Section 4). For example, why do certain parameters have different responses to different optimization methods? What are the limitations of multiple-site optimizations? Adding some discussion could really solidify the take home points of this paper and provide relevance to other land surface models and parameter optimization studies.

Answer: The major factors influencing the parameter estimates are related to the technical implementation of the optimization methods – whereas the gradient-based method mostly looks for the optimal parameter set in the vicinity of the prior parameter values, the random search algorithm may jump to a completely different parameter state in one step. This is the main reason why we obtain significant differences in the estimated parameters between L-BFGS-B (BFGS) and Genetic Algorithm (GA), no matter which case, single-site (SS) or multi-site (MS) is selected. The differences are pronounced for specific parameters such as C\_T\_opt, LAI\_max, SLA, L\_age\_crit, K\_lai\_happy, K\_GR and K\_rsoil mainly. In the particular case of using pseudo-data, the GA manages to find the true values for these parameters much more precisely than the BFGS algorithm. In most cases, with real data we also see differences between BFGS and GA for these parameters, and we can thus speculate that the GA would provide more optimal posterior estimates in this case as well.

On the other side, if we compare SS vs MS, we do not observe specific patterns in the posterior parameter values, but the range of parameter values obtained with multiple first guesses are significantly lower for MS than for SS. This comes from the fact that for SS cases each site is optimized separately, so we can end up with a parameter value that is highly specific of each site, whereas for MS cases we optimize all the sites together, so the final estimate has less variability for the multi-site optimization. This is illustrated more specifically by the parameters Kw\_root, Ct\_sen, Q10, HRha, HRhb and Kz0.

Concerning the limitations of the multiple-site optimization, we would like to raise the following points. First as discussed in the paper the benefit of assimilating multiple sites of a given PFT follows from the need to neglect site peculiarities and to find an optimal set of parameters describing the PFT in general. However, the optimization usually does not work efficiently (i.e. does not lead to a large decrease of the cost function) if the different sites have very different

behaviors in terms of carbon/water cycle responses to climate forcing. This informs us on the need to reconsider the PFT geographical description (with possible further regional split). This is slightly the case for TropEBF and C3 grass. Additionally, the use of multiple-site optimization requires more computing time and is slightly more complicated to set up with the need to have coherent observation errors between the sites, i.e. with no site that dominates the cost function because of a too low error (measurement and model errors grouped in the R term) and thus prevents the optimization to fit all sites together.

Overall, we agree that these two points were not detailed enough in the manuscript and we have thus included the points discussed above in the main text of the paper (in section 3.2.2 – second and fourth paragraphs for the parameters discussion and at the end of the section 3.1.2 for the limitations of the multi-site optimizations).

**Referee: Though the focus is parametric uncertainty, I think the paper would also benefit from a brief discussion of model structural uncertainty, with relevant details specific to ORCHIDEE. This addition would provide useful context for discussing the results (e.g., Page 10, Line 25).**

Answer: We agree that model structural uncertainties are also a critical part of any data assimilation experiment, but they are rather difficult to assess properly. However, from the existing knowledge on the different processes that control the land surface carbon, water and energy budgets we can list potential missing processes in ORCHIDEE that may have a direct impact on the parameter retrieval. For instance, the version used in this study still lacks a description of the nitrogen cycle and its potential limitation on photosynthesis (in the context of increasing atmospheric CO2), which may bias the retrieval of Vc\_max parameter. We also do not describe properly forest stand and canopy structure (forest gap, age dependent effects, etc.), which is a limitation on the computation of the absorbed light for photosynthesis and of the turbulent fluxes exchanged with the atmosphere. The main risk is indeed to over-tune some parameters for wrong reasons (i.e., because of missing or incorrect process description) – that some of the structural error will be aliased onto the model parameters. However, this is not the focus of this study so we do not go into too much detail here. It is described in depth in MacBean et al. (2016).

Overall we do agree a brief discussion of this issue would be useful in the text, so we have slightly extended the model description (section 2.1), to mention the importance of model structural uncertainties and listed potential effects of missing processes on the parameter retrieval.

**Referee: I think it would be useful to include some background on choice of model parameters and how their sensitivity was assessed, as this is a key step to narrowing the parameter space.**

Answer: The choice of the model parameters was done based on sensitivity tests of the data used in the optimization (net CO2 ecosystem exchange (NEE) and latent heat (LE) fluxes) with respect to carbon and water cycle related ORCHIDEE parameters. This was done in previous works, as already cited in the paper and based on the so-called Morris method (Morris, 1991). As nothing new was introduced in this study, we had limited the background description of this choice in section 2.2. However, we agree that more information on the subject is useful, so the following sentences are added in section 2.2 together with the new reference:

"Among all ORCHIDEE parameters we selected the ones that primarily drive net CO2 ecosystem exchange (NEE) and latent heat fluxes (LE) variations on synoptic to seasonal timescales, excluding those impacting preferentially decadal time scales (i.e., like tree mortality). A preliminary parameter sensitivity analysis was conducted, as in Kuppel et al. (2012), based on the "one-at-a-time" Morris method (Morris, 1991), and we restricted the selection to the 28 most influencing parameters controlling photosynthesis, respiration fluxes, leaf phenology and evapotranspiration."

**Referee: Minor note, but it would be preferable to have the line numbers continuously increasing throughout the document so identifying page numbers in the specific comments is not necessary.**

Answer: We agree that using a continuous line numbering would facilitate the referencing, we will follow this advice in the next revision.

**Referee: Page 3, Line 2: "probability distribution function"**

Answer: The typo is corrected.

**Referee: Page 3, Line 6: What are the limitations of the Gaussian assumption? Could some parameters have different PDFs?**

Answer: The currently implemented data assimilation technique relies on the assumption that the errors on both the parameters and the observations have Gaussian PDF. In this case, the resolution of the inverse problem, following a Bayesian framework, is equivalent to the minimization of a quadratic cost function. This Gaussian hypothesis significantly simplifies the interpretation of the minimum of the quadratic cost-function (i.e. being the mean of the posterior parameter PDF). If some parameters would have other PDF, the L-BFGS-B minimization procedure would not provide a meaningful value to describe the posterior PDF. However, for the GA this restriction does not hold, as we could use the ensemble of model trajectories to describe the posterior PDF. Additionally, the Gaussian hypothesis is also central to calculate a posterior parameter uncertainty that fully describes (together with the mean value) the shape of the PDF, and this hypothesis also allows to compute the posterior uncertainty with a simple matrix formulation (see for instance Tarantola, 2005). Such hypothesis is used in many inversion problems.

However, we agree that non-Gaussianity may be the case for some parameters and that it could thus partially bias the overall parameter optimization with Gaussian assumption. It is nevertheless out of the scope of this paper to investigate non Gaussian errors. We have examined the issues that may arise when assuming Gaussian PDFs in MacBean et al. (2016). Moreover, in the case of ORCHIDEE we have shown in an earlier study (Santaren et al., 2007) that most parameter errors follow Gaussian distributions. We have inserted these points at the end of section 2.4.1:

"Note that using non-Gaussian errors would significantly complicate the application of one class (gradient-based) and is thus out of the scope of this study. MacBean et al. (2016) examined the issues that may arise when using Gaussian assumptions in gradient-based minimisation algorithms; however, they found that the algorithm used in this study could account for quasi non-linearity. Moreover, in the case of ORCHIDEE we have shown in an earlier study (Santaren et al., 2007) that most parameter errors follow Gaussian distributions."

**Referee: Page 3, Line 19: What defines "excessive" here? Can you give examples of the number of parameters explored in these studies, and how they compare to the dimensionality of your problem?**

Answer: In the cited study (Chorin and Morzfeld, 2013), it was shown that the effective problem dimension (defined as the Frobenius norm of the steady state posterior covariance) can remain moderate for realistic models even when the state dimension (i.e. the number of parameter in our case) is large (asymptotically infinite). The precise value of the excessive effective dimension varies from one problem to the other and depends on the level of accuracy required.

However, obviously the effective dimension has to remain bounded. In our study the dimensionality of the problem is limited to the few tens of parameters and it can be considered to be small as compared to the cited study (going up to a thousand), which supports the main idea that the numerical data assimilation can be successful.

We slightly modified the text to include the definition of the problem dimension as it is meant in the cited study and changed the phrasing "not excessive" to "finite" for a clearer readability:

"With idealised models, Chorin and Morzfeld (2013) have shown that the assimilation can be optimal with particle filters or variational methods, under the condition that the effective dimension of the problem (defined as the Frobenius norm of the steady state posterior covariance) is finite".

**Referee: Page 3, Line 33: L-BFGS\_B should be L-BFGS-B?**

Answer: The typo is corrected.

**Referee: Page 4, Line 9: Word choice "exploited" could be changed to "utilized".** Answer: The word "exploited" is changed to the word "used".

**Referee: Page 4, Line 13: Change to "a few".**

Answer: The missing word is added.

**Referee: Page 4, Line 16: ORCHIDEE should be defined at first mention (Page 3, Line 23 and in abstract).**

Answer: The ORCHIDEE transcription as it exists (ORganizing Carbon and Hydrology In Dynamic Ecosystems) is more a play of words, than the real meaningful definition. So, we decide to mention it only in the devoted section (section 2.1) and not in the abstract and earlier mentioning of the model in order not to be misleading.

**Referee: Page 4, Line 19: Why "possibly"? Has the use of ORCHIDEE on thousand-year timescales not been proven?**

Answer: Indeed, the use of ORCHIDEE model is proven on the long timescales basis. The word "possibly" was only used to show that it is "possible" to run the model on such timescale. The word is now removed for a smoother readability.

**Referee: Page 5, Line 5ff: A few lines about the choice of parameter ranges and sensitivity assessment would be useful here (even if in the supplemental to go along with Table S1).**

Answer: The ranges of variation for the parameter values have been assigned based on literature analysis and parameter database such as the TRY database (http://www.try-db.org) as well as following expert knowledge of the model equations. We added this point in the revised manuscript.

For the sensitivity assessment, we agree that additional information is useful as already discussed/provided in the answer to the third reviewer comment.

**Referee: Page 5, Line 13: Repetitive here to again mention land use change in parentheses.**

Answer: The typo is corrected, the first occurrence is deleted and the one at the end of the sentence is kept.

**Referee: Page 5, Line 20: How useful is 1 year of station data? Only one observed seasonal cycle, especially relevant as optimization is on seasonal/annual time scales. Relevant also at Page 10, Line 23.**

Answer: Even only one year of data already provides valuable information on the 'main' seasonal cycle and important information on the ecosystem response to synoptic weather events. In general, we thus tried to keep as much data as possible, even if we had only one year of data for a specific site. However, in the case of the multiple sites optimizations (MS), this may lead to some representativeness issues, with long-record sites dominating the cost function and the overall optimization. We have faced this problem and for some sites with a small amount of data the optimization could led to a degradation of the model – data fit in few MS cases. Although using sites of similar record length would be optimal, we believe that keeping short record sites is still crucial to account for the diversity of ecosystem within a given PFT.

We did not revise the text as this point is already mentioned in page 10 (section on multi-site optimization).

**Referee: Page 5, Line 27: Please add a sentence explaining why you would expect the Bowen ratio to be constant.**

Answer: The correction of the energy balance closure is a difficult task and experts in eddycovariance flux measurement have not put into evidence that one of the turbulent heat fluxes (latent or sensible) is on average more impacted than the other one. Using a constant Bowen ratio is thus a conservative and natural choice that is applied in most studies (Lasslop et al. 2008, Twine et al. 2000). We slightly modified the text to include this point:

"Where possible, the LE fluxes have been corrected to close the energy balance, using measurements of the ground heat flux (G) and keeping a constant Bowen ratio to update the latent and sensible heat fluxes (i.e., conservative choice without strong evidences that one turbulent flux may be more impacted than the other one; Twine et al., 2000)."

**Referee: Page 6, Line 19: Should be Tables S3-S4.**

Answer: Corrected.

**Referee: Page 6, Line 23: L-BFGS-B acronym should be defined at first mention, in introduction Page 3, Line 30 (and abstract).**

Answer: Corrected in the text and added in the abstract.

**Referee: Page 7, Line 1: Change "to threshold" to "with threshold".** Answer: Corrected.

**Referee: Page 9, Line 13: Would be nice to include the equivalent of Figure 1 for LE flux. Figure S1 has it broken out by PFT but not a summary figure.**

Answer: We chose initially not to include the equivalent of Figure 1 for the LE flux as we mainly focus on the carbon fluxes in this paper. However, we now follow the reviewer's suggestion and we have added it into the Appendix as the ending part of the multi-panel in Figure S1.

**Referee: Page 10, Line 25: This is a key point – how do model structural uncertainties get in the way of multiple site optimizations? What are the limits to finding an optimal parameter set across multiple sites? (Also "degrade" should be "degrades"?)**

Answer: As already discussed above in the response to the general comments, we agree that model structural uncertainties are crucial in the optimization process but difficult to assess. Multiple sites optimizations are likely to reveal more directly the impact of structural uncertainties

as the optimization will not be able to fit simultaneously all data streams, while in a single site optimization some parameter changes may more easily compensate for model structural errors. However, given that the primary objective of the optimization of ORCHIDEE is to improve the model for large-scale applications (regional to global), the use of a multiple sites optimization is the only way to account for the 'within PFT' diversity. The limits arise when the objective is to study the response of a specific ecosystem (and not a generic PFT) to external drivers as the multiple site parameter set might be sub-optimal for the particular ecosystem. We have thus included few sentences in the text to highlight these issues (see end of first paragraph in section 3.1.2).

The mentioned typo is corrected.

**Referee: Section 3.1.3: This section could be moved earlier in the paper as it is referenced in earlier parts of the results. Overall the flow of the results section could be improved.**

Answer: We agree that this section could potentially be moved upfront. However, it would then somehow hide the first order message of the paper linked to the comparative performances of the two algorithms, gradient-based versus Genetic. We thus propose to keep the section where it is but to improve the overall flow of the results section. We have i) dropped the introduction in section 3.1.3 as it was redundant with the justification for multiple first-guess tests in the method section, ii) improved slightly the method, section 2.5, to better justify the use of several first-guess tests and iii) added a sentence at the beginning of the results section (first paragraph in 3.1) to explain the flow of results and the different sub-sections.

**Referee: Page 11, Line 32: First use of "SG" – replace with S\_Genetic? That is the abbreviation used in other parts of the text.**

Answer: The typo is corrected.

**Referee: Page 12, Line 27: Why does the pseudo-observation experiment perform poorly for Z\_crit, litter? Why does it perform better for other parameters?**

Answer: The poor performance for Z\_crit, litter is likely to be related to the relatively low sensitivity of the model outputs (NEE and LE) with respect to that parameter. This is by comparison to the other parameters. We have mentioned this reason in the text.

**Referee: Page 13, Line 8ff: What drives different parameters to respond better or worse to different optimization methods?**

Answer: Overall the main reasons that drive the differences in the parameter response to the different optimization methods are linked to the sensitivity of the chosen model output to each parameter, the prior parameter errors and error correlations and the overall shape of the cost function with respect to parameter sensitivity at any point in parameter space. The two algorithms explore parameter space in very different ways, therefore they deal with complicating issues (i.e. the existence of local minima) in different ways. We do have model equifinality in our results as the result of parameter error correlations and a lack of prior constraint (high prior uncertainty). As a result, we will not necessarily obtain the same posterior parameter vector but will achieve the same reduction in model-data misfit. The GA in particular is a random walk, therefore it is possible that it converges on a different parameter vector than the BFGS but with the same reduction in model-data misfit.

However it is difficult to investigate more deeply why one particular parameter is more sensitive to the Genetic or the Gradient-based algorithms in the context and scope of this paper. We thus believe that a general explanation of the reasons underlying these differences is the

appropriate level of detail. We have thus only slightly changed the text in section 3.2.1 to better highlight the reasons of these differences.

**Referee: Page 13, Line 12: The error in Z\_crit, litter was mentioned on the previous page as 29%, please clarify.**

Answer: It was rounded in the second case. Indeed, it makes no sense to put the numbers again here, where we compare the differences between the methods. The sentence has been changed to: "A few parameters are not well estimated by both methods, like F\_stressh, HR\_Hb and Z\_crit\_litter, having the largest difference with respect to the true value."

**Referee: Page 13, Lines 27, 33: Same question as previous section; what drives different responses in different parameters?**

Answer: The response is similar to that for the case of the pseudo-data experiment (previous section). We have thus referred in this section to the previous one for the explanation of the causes of the different responses between the different parameters.

**Referee: Page 14, Line 20ff: Some grammar issues in the bullet points, and throughout this section (e.g., Page 15, Lines 5, 16 and 23).**

Answer: The bullet point text together with the Summary section has been proofread.

**Referee: Page 15, Line 18: I think you should mention this point in Section 2.4.3.**

Answer: Indeed, this could be considered as an additional feature of the minimization algorithm. The following sentence is added in the end of Section 2.4.3:

"Contrary to gradient-based methods, random search algorithms allow to use any form of probability distribution functions for the observation and parameter uncertainties, and thus to use non-Gaussian PDFs."

**Referee: Figure 1: In legend, use dots instead of lines to help guide the reader.** Answer: Lines are changed to dots on the figure.

**Referee: Figure 3: Add error bars here, the uncertainty on the RMSD reduction is referenced in the text (e.g., Page 11, Line 28).**

Answer: It's not the uncertainty on the RMSD reduction that is referenced in the text (Page 11, Line 28), but the full range of variation for the single-site Genetic optimization which goes from 50.5% with only one first guess to 55% with 16 first guesses (see Fig. 3, red line). As the notation 52±2% happened to be misleading, we change the text and now refer to the range of variation, 50.5–55.0%. Besides, there is no obvious uncertainty that we can add on Figure 3, because the RMSD reduction that we plot corresponds to the average of the "maximum RMSD reductions" obtained across all possible groups of N first guesses (X axis). Such measure is thus not directly a stochastic variable. Although we could add for each number of first guesses (X axis) the standard deviation across all "maximum RMSD reductions", it does not change much with the number of first-guesses, so we choose not to add this information as it would complicate the description of the figure and will not add much to the overall message.

**Referee: Table S1: Missing units for parameters, as applicable.**

Answer: The missing units for parameters are added.

**Referee: Tables S3, S4: PFTs are numbered but not specified by name.**

Answer: PFT names are added in the tables.

**Referee: Figure S1: Have the legend in one panel and get rid of them elsewhere, they are just distracting/overlapping data points.**

Answer: The legend is now kept only on the first panel and dropped for the others.

**Referee: Figures S2, S3: Where in the main text are these figures referenced?**

Answer: These figures were indeed added to support the interpretation of the results but not properly referenced in the text. The references have been added.

**Referee: Figure S3: Add error bars, same comment as Figure 3.**

Answer: We choose not to add any error bars following the response above for Figure 3.

**Referee : Figure S4: If some parameters do not apply to certain PFTs (e.g., K\_pheno,crit), why are they optimized for that PFT? Is this an error in Table 1 and/or Table S1?**

Answer: This is a mistake in Table S1 and Figures S4. Indeed, there are two specific parameters that apply only to selected PFTs (K\_pheno,crit applies to deciduous PFTs only and thus TempDBF, BorDBF, C3 grass; C\_t,sen applies to TempDBF and BorDBF). The corresponding table/figures are corrected.

Best regards, Vladislav Bastrikov

"Note that this study does not aim to provide an exhaustive examination of all methods belonging to both classes of inversion algorithms (as previously described), nor do we presume to have chosen the best method belonging to each class. We simply choose to test two methods belonging to each class that have already been used to estimate parameters of the ORCHIDEE model. A further examination of the benefits of all methods would be beneficial to the LSM and DA community, but is outside the scope of this study."

**Referee: P2, L12, L13, L15: These error sources (vegetation and soil spatial information versus parameter values in model) are partially the same thing? Please clarify.**

Answer: We believe that these error sources relate to different aspects of the model. The vegetation and soil spatial information represents the so-called forcing data sets (like the meteorology); they correspond to global maps derived partly from satellite earth observation missions and contain the necessary information to cluster the different types of soils and vegetation all over the globe. The parameter values correspond to specific and chosen formulations of the different processes controlling the carbon, water and energy budgets in the model. They are thus internal to the model while the forcing data are external and may easily vary depending on the region where the model is applied. It is thus helpful to distinguish these different error sources and to classify them into separate groups.

We have thus only slightly changed the text to be more precise.

**Referee: P5, L7: Could you provide more details? How is the sensitivity study conducted? Reference?**

Answer: The choice of the model parameters was done based on sensitivity tests of the data used in optimization (net CO2 ecosystem exchange (NEE) and latent heat (LE) fluxes) with respect to all related ORCHIDEE parameters. This was done in previous works, as already cited in the paper and based on the so-called Morris method (Morris, 1991), which ranks the variability of

"elementary effects" of the sampled parameters with respect to their impact on the model outputs. This information together with the new reference has been added to the reviewed text.

**Referee: P5, L22: Can you provide more details? How many sites were disregarded?**

Answer: Based on an original list of 252 sites from the La Thuile dataset (Baldocchi, 2001), we conducted a screening as described in the paper resulting in the selection of the 78 sites used in this study. Thus, we have disregarded around 70% of the sites. The total number of the sites in the original database has been added to the reviewed text. Additional reference to the article devoted to the PFT refinement subject is also added: "Note that Peaucelle et al. (in review) explored with the same model how to account for plant functional trait variability, refining the PFT distribution".

**Referee: P6, L31: How automatic was the automatic differentiation? Can you provide more details on the additional coding and time which was required?**

Answer: The differentiation has been generated by the TAF automatic differentiation tool from the FastOpt company (see http://www.fastopt.com/). However, the success of automatic differentiation largely depends on the cleanliness of the model code and to a certain extent on the structure of the code. Our group had spent a large effort on cleaning and making the initial ORCHIDEE model code, suitable for the TAF software. Note that a specific document with recommended coding guidelines has been built as the result of this work. Additionally, some input/output peace of code was also inserted to handle the tangent linear variables that are differentiated through TAF. Overall, this work required a strong investment of one software engineer and it took us around two years to have a working tangent linear model.

We thus added in the text one sentence to resume the committed effort: "For ORCHIDEE the corresponding TL model has been derived with the automatic differentiation tool TAF (Transformation of Algorithms in Fortran; see Giering et al., 2005), following code cleaning and structural adjustments (without changing the physics) to allow the use of TAF (a significant effort that took around two years)"

**Referee: P8, Eq. 2: How critical is the Gaussian assumption? Non-Gaussianity of parameters can be expected.**

Answer: The currently implemented data assimilation technique relies on the assumption that the errors on both the parameters and the observations have Gaussian PDF. In this case, the resolution of the inverse problem, following a Bayesian framework, is equivalent to the minimization of a quadratic cost function. This Gaussian hypothesis significantly simplifies the interpretation of the minimum of the quadratic cost-function (i.e. being the mean of the posterior parameter PDF). If some parameters would have other PDF, the L-BFGS-B minimization procedure would not provide a meaningful value to describe the posterior PDF. However, for the GA this restriction does not hold, as we could use the ensemble of model trajectories to describe the posterior PDF. Additionally, the Gaussian hypothesis is also central to calculate a posterior parameter uncertainty that fully describes (together with the mean value) the shape of the PDF, and this hypothesis also allows to compute the posterior uncertainty with a simple matrix formulation (see for instance Tarantola, 2005). Such hypothesis is used in many inversion problems.

However, we agree that non-Gaussianity may be the case for some parameters and that it could thus partially bias the overall parameter optimization with Gaussian assumption. It is nevertheless out of the scope of this paper to investigate non Gaussian errors. Moreover, in the case of ORCHIDEE we have shown in an earlier study (Santaren et al., 2007) that most parameter errors follow Gaussian distributions. We have inserted these points at the end of section 2.4.1:

"Note that using non-Gaussian errors would significantly complicate the application of one class (gradient-based) and is thus out of the scope of this study and that Santaren et al. (2007) have shown with a previous version of ORCHIDEE that most parameter errors follow Gaussian distributions."

Referee : P14, L30: How do you know whether this number (5) is not case dependent? In case of multiple sites, many tests could be carried out, as the parameters need to be determined just once (per PFT of course). Why would one need to impose restrictions, and would it not be better to use a larger number of initial guesses for cases with parameter estimation for multiple sites?

Answer: We agree with the reviewer that the larger the number of first-guess tests is, the more robust the results will be. However, this has to put in regards to the computing time that is increasing proportionally to the number of first-guesses. Note that the computing time does not vary substantially between the single and the multi-site cases as even for the multi-site case the model has to be run at all sites for each iteration. This chosen number (5) comes from a first order analysis of the results presented in Figure 3 with our particular model and set of observations. It should be seen as a first order and prior suggestion. For any other model and parameter inversion exercise it could of course be different. Note that this summary point is primarily to stress the fact that using only one first-guess inversion is very risky with a gradient-based method.

Overall, we agree that this statement needs to be put into a more general context and perspective. We have thus changed the text to include the above elements.

**Referee:**

P1, L26: trapped instead of trap.
P10, L25: degrades instead of degrade
P10, L29: skip "a"
P10, L32: "reductions" instead of "reduction".
P11, L11: "at the same level as" instead of "at the same level than".
P11, L25: "maxima" instead of "maximums". The results are shown in?
P11, L31: "as" instead of "than".
P13, L5: "minima" instead of "minimum".
P13, L28: Change to: "the most constraint ones"
P14, L17: "dependent"
P14, L30: change to: "ensures".
P15, L5: change to "decreased".

Answer: All mentioned typos are corrected in the text. We thank very much the referee for the thorough and attentive reading.

Best regards, Vladislav Bastrikov

**Land surface model parameter optimisation using in-situ flux data: comparison of gradient-based versus random search algorithms (a case study using ORCHIDEE v1.9.5.2)**

Vladislav Bastrikov1,2, Natasha MacBean1\*, Cédric Bacour3, Diego Santaren1, Sylvain Kuppel4 and Philippe Peylin1

1Laboratoire des Sciences du Climat et de l'Environnement, UMR 8212 CEA-CNRS-UVSQ, 91191 Gif-sur-Yvette, France 2Institute of Industrial Ecology UB RAS, 620219 Ekaterinburg, Russia 3Noveltis, 153 rue du Lac, 31670 Labège, France

[revised manuscript text omitted]

- 65 simultaneously –usually at the level of PFT (Groenendijk et al., 2011; Kuppel et al., 2014; Raoult et al., 2016). These studies have employed various data assimilation (DA) techniques inversion approaches, but they all rely on a Bayesian framework that allows the update of a priori information on the parameters (i.e., the parameter prior <del>probably probability</del> distribution functions – PDFs) with using the new information contained in the observations, leading to which results in the posterior PDFs. While some methods do not require any assumption as to the shape of the different PDFs (see for example
- Van Oijen et al., 2005and Post et al., 2017, who use Markov Chain Monte Carlo (MCMC) methods), most of themmany inversion methods use Gaussian error assumptions (for both parameters and observations), which simplifies the characterisation of the posterior parameter PDFs and leads to the minimisation of a cost function (Dietze et al., 2013). Our study uses such Gaussian hypothesis.

The optimisation requires the minimisation of a cost function that describes the difference between the model and the

- observations, taking into account their respective uncertainties. There is a wide variety of mathematical algorithms that can be used to explore the parameter space (in the case of DA for parameter estimation), and therefore for locateing the minimum of a cost function, and tThese methods can be broadly divided into two classes;; i) local gradient-based methods (i.e., gradient-descent or conjugate gradient algorithms for example) also referred to as variational, or 4D-Var, methods; and ii) global random search method (i.e., Markov Chain Monte Carlo (MCMC), genetic and particle filter algorithms for the
- 80 most common onesare commonly used examples). There are various advantages and disadvantages for the different algorithms of each class, depending on the complexity of the model being optimised complexity (mainly the degree of model non-linearity), the simulation computational time of each model simulations, and the number of parameters to be optimised. Random search methods (as used in Van Oijen et al., 2005; Richardson et al., 2010; and Post et al., 2017 for example) have been proven to be efficient in finding the correct parameter values with highly non-linear models but at the expense of
- 85 potentially prohibitive computing cost. On the other hand, gradient-based methods (as used in Kuppel et al., 2014; Raoult et al., 2016; Pinnington et al., 2016; and Schürmann et al., 2016 for example) are more prone to getting stuck in local minima and, furthermore, they require the complex calculation of the gradient of the cost function with respect to all parameters. With idealised models, Chorin and Morzfeld (2013) have shown that the assimilation can be optimal with particle filters or variational methods, under the condition that the effective dimension of the problem (defined as the Frobenius norm of the
- 90 steady state posterior covariance) is not excessive finite. The choice of the minimisation method was shown to have little impact on the overall optimisation efficiency for relatively simple ecosystem models (Trudinger et al., 2007; Fox et al., 2009). Ziehn et al. (2012) showed similar performances between a MCMC and a variational method in terms of locating the minimum of the cost function for the BETHY ecosystem model (using atmospheric  $CO_2$  data as constraint), except for the difference in computational time (much longer using MCMC). However, with the more complex ORCHIDEE LSM,
- 95 Santaren et al. (2014) showed that the genetic algorithm was superior to a gradient-based method in reducing the cost-function to the correct minimum at one FLUXNET site (using water and carbon fluxes as constraint), as the gradient method
  was often stuck in local minima. Our study further expands their analysis to multiple-sites and across multiple PFTs.

The overall objective of this study therefore is to investigate the trade-off between these two classes of method in terms of their computational efficiency versus their ability to constrain the parameters to the correct value. To achieve this goal, we

- 100 performed experiments using i) a global state of the art process-based land surface model (ORCHIDEE, Krinner et al. 2005), ii) a large ensemble of  $CO_2$  and water flux measurement at FluxNet sites and iii) one particular variant for each class of method, the gradient-based L-BFGS-B algorithm (limited memory Broyden-Fletcher-Goldfarb-Shanno algorithm with bound constraints, Byrd et al., 1995) and a genetic algorithm (GA) for the random search method. Note that this study does not aim to provide an exhaustive examination of all methods belonging to both classes of inversion algorithms (as previously
- 105 described), nor do we presume to have chosen the best method belonging to each class. We simply choose to test the two abovementioned methods belonging to each class (L-BFGS-B and GA) that have already been used to estimate parameters of the ORCHIDEE model (Kuppel et al., 2014; Santaren et al., 2014; MacBean et al., 2015; Bacour et al., 2015; Peylin et al., 2016). A further examination of the benefits of all methods would be beneficial to the LSM and DA community, but is outside the scope of this study.
- 110 The key questions that we investigate are the following:
  - Does the random search algorithm (GA) result in a lower spread inposterior cost function minimum value than the L-BFGS\_B gradient-based method (BFGS) for the single site (SS) tests?
  - ii) Are the differences in cost function minimum between the GA and BFGS methods smaller for multi-site (MS) optimisations than for single sites (SS), following a potential smoothing of the cost function shape with a greater number of observations?
  - iii) What is the spread of the posterior parameter values with the BFGS and the GA methods when performing the same tests with multiple first guesses in parameter space? Is the spread different between the two methods? aAnd how many first guesses are needed for each method to improve the location of the cost function minimum?
  - iv) Does one method result in a closer approximation of the true posterior parameter value in the pseudo-observation tests, for both the SS and MS experiments?

Section 2 presents the ORCHIDEE model and its configuration, describes the observational data exploited-used in the study and gives a detailed explanation of the two data assimilation methods as well as the setup of the experiments. A comparison of the performances of the two optimisation methods and of the differences between the single and multiple sites approaches is done-detailed in Section 3, including the analysis of model–data misfit (Sect. 3.1) and estimated parameter values (Sect. 3.2). The last section summarises the main findings and proposesvides a few perspectives to how this workmay be extended in the future.

120

125

115

**2 Materials and methods**

**2.1 The ORCHIDEE land surface model**

We used the process-based land surface model ORCHIDEE (ORganizing Carbon and Hydrology In Dynamic Ecosystems)

- 130 version 1.9.5.2 that has been developed at the IPSL (Institut Pierre Simon Laplace, France). It simulates the carbon, water and energy exchanges between the land surface and the atmosphere (Krinner et al., 2005). The model is applicable over a wide range of spatial scales (from "grid-point" to regional or global) and covers timescales from 30 minutes to possibly thousands of years. ORCHIDEE may be deployed as a stand-alone terrestrial biosphere model using meteorological forcing (observational data or model reanalysis) or as part of the IPSL Earth System Model (Dufresne et al., 2013) used for climate
- 135 predictions. The version that we use corresponds to the one used for the CMIP5 simulations for the IPCC 5th Assessment Report.

The hydrological and photosynthesis processes as well as the energy balance are treated at a half-hourly time step, while the slower components linked to the carbon allocation within the plants, the autotrophic respiration, leaf onset and senescence, plant mortality and soil organic matter decomposition are processed at a daily time step. The soil hydrology model used in

- 140 this study corresponds to a simple double-bucket scheme (Ducoudré et al., 1993). The land surface is represented by 13 plant functional types (PFT), including bare soil, that can co-exist in any grid cell. Except for the phenology, the processes are described by generic equations but with different parameters that are PFT-specific. For the main model equations, the reader is referred to numerous previous publications (e.g. Krinner et al., 2005) that are reported on the ORCHIDEE web-site (http://orchidee.ipsl.fr).
- 145 For this study we applied the model at site level (see section 2.3) using gap-filled half-hourly meteorological data measured at each site (air temperature, humidity, pressure, wind speed, rainfall and snowfall rates, shortwave and longwave incoming radiation; see Vuichard et al. 2015). The soil carbon pools have been brought to equilibrium (spin-up procedure) by cycling the available meteorological forcing over several millennia (with present day CO2 concentration) in order to ensure that the long-term net carbon flux is close to zero.
- 150 It is important to note that model structural errors, due to missing or poorly described processes may directly impact the parameter retrieval. For example, the ORCHIDEE version used in this study lacks a description of the nitrogen cycle and its potential limitation on photosynthesis (in the context of increasing atmospheric CO2) and a proper description of forest stand and canopy structure (forest gap, age dependent effects, etc.) which is a limitation on the computation of the absorbed light for photosynthesis and of the turbulent fluxes exchanged with the atmosphere. These missing processes (that will integrate
- 155 future model versions) may lead to over-tuning some parameters and potentially decrease the predictive skill of the model. Nonetheless, it does not change the main outcomes of the study, which refer to optimisation method comparison rather than effective model improvement.

**2.2 Optimised parameters**

The ORCHIDEE parameters selected for the optimisation are described in Table 1 along with their default values. They have

ones that primarily drive net CO2 ecosystem exchange (NEE) and latent heat fluxes (LE) variations on synoptic to seasonal

[revised manuscript text omitted]

The GA does not require any gradient calculations; therefore, one chromosome rank estimate requires one standard model run and the total computational time for the GA assimilation for one site equals to 1200 runs. Contrary to gradient-based methods, random search algorithms allow to use any form of probability distribution functions for the observation and parameter uncertainties, and thus to use non-Gaussian PDFs.

**270 2.4.4 Posterior uncertainty**

Assuming Gaussian prior errors and linearity of the model in the vicinity of the solution, the posterior error covariance matrix of the parameters, **A**, can be approximated by:

$$\mathbf{A} = [\mathbf{H}^{\mathrm{T}} \mathbf{R}^{-1} \mathbf{H} + \mathbf{P}_{b}^{-1}]^{-1},$$
(2)

[revised manuscript text omitted]

- 330 lower with the genetic algorithm the  $S_{Genetic}$  method is only slightly disadvantageous for 3 sites out of 78, although the spread across the 16 first guesses remains low (<10%). Overall, the mean spread of the RMSD reduction for the  $S_{Genetic}$  case is around 10%. This clearly indicates that the genetic algorithm, following the set up described in section 2.4.3, is able to find nearly the same minimum of the cost function independently from the choice of the first guess parameters (with a significant improvement of the model-data misfit, see Figure 1). On the contrary, the results of the  $S_{BFGS}$  method are highly
- dependent on the first guess model parameters, with a spread above 22% for half of the sites. Finally, if we compare the best achieved RMSD reduction within the 16 first guess tests (see Tables S3 and S4 in the Supplement), we obtain similar performances for most of the sites of 5 PFTs out of 7, excluding TropEBF and BorDBF, where the SGenetic method still produces better results than the SBFGS method.
- With the gradient-based method ( $S_{BFGS}$ ), using the standard ORCHIDEE model parameters as a first guess does not guarantee an optimal reduction of the cost function – the corresponding posterior RMSD could be either the lowest or the highest one of the 16 tests (these cases are shown by a circle, as opposed to a cross, in Figure S1 that displays the posterior RMSD for all sites and all 16 tests). Although we have used the same random parameter sets for each site of a given PFT, the "optimal" first-guess parameter set (i.e., the one providing the largest cost function reduction) differs between sites. It indicates that the shape of the cost function varies between sites and that there is no optimal prior first guess with the SBFGS
- 345 gradient method, which is prone to getting caught in local-minima if the assumption of model quasi-linearity is not met.

Overall, this highlights one weakness of gradient-based methods and the need to perform several independent assimilations starting from different first-guess parameter values (see section 3.1.4).

**3.1.2 Multi-site optimisation: comparative performances of the methods (MBFGS vs MGenetic)**

- The reduction in NEE RMSD for the multiple site optimisations (MBFGS for the gradient-based method and MGenetic for the 350 Genetic algorithm) is illustrated in the lower panels of Figure 2. First, the multi-site optimisations provide lower RMSD reduction than the single-site optimisations (lower vs upper panels of Figure 2). This is the trade-off between fitting a specific site versus fitting an ensemble of sites representing more accurately the diversity of plant, soil and climate for a given PFT. We then notice that for few sites the RMSD is increasing after the optimisation (i.e. negative value of the RMSD reduction): 1 site for TropEBF and C3grass PFTs and 3 sites for TempENF and BorENF. On average these sites have only one or two years of data with large prior observation errors and thus a smaller weight in the cost function compared to the other sites of the same PFT. This could thus explain that the optimisation degrades the model-data fit at these sites but it could also indicate that there is no optimal parameter set improving the model-data fit at all sites, suggesting the need to refine the PFT description and/or to improve the model structure (see a list of potential impacts of missing processes in the
- ORCHIDEE model in section 2.1). Note that multi-site optimisations are likely to be more impacted by model structural
   errors than single site optimisation, as these errors may have different impacts at each site. Limits arise when the objective is to study the response of a specific ecosystem (and not a generic PFT) to external drivers as the multi-site parameter set might be sub-optimal.

The performance of the two methods differs between the PFTs. For TropEBF, TempENF and C3grass PFTs the genetic algorithm still provides on average significantly larger RMSD reduction than the gradient method, with a-ratios between the

- two of 2.0, 1.8 and 1.4, respectively. For the other four PFTs both methods show much smaller differences and for TempDBF the average RMSD reduction is exactly the same. Overall, at 50 sites (from 78) the MBFGS method provides
  performances comparable to the MGenetic method (RMSD reductions differ by less than 25%) or even slightly better (see Tables S3/S4 in the Supplement for all numbers) and the mean additional RMSD reduction for the MGenetic method is only 6%.
- 370 The spread across random first guesses between the two methods is much more comparable than for the single-site optimisations. The  $R_{90}$  values for the gradient-based method are only considerably larger for half of the sites than the genetic algorithm method (with values up to 2 times larger), except for two PFTs (TempENF and TempDBF) where they are similar on average (Figure 2, lower right panel). This suggests that increasing the number of observations and/or capturing a greater range of sensitivity to the parameters acts to linearise or smooth the cost function, thereby ensuring that the gradient-based

method does not get as caught in local minima as in the SS optimisations.

375

The last point to notice is that using the standard ORCHIDEE parameter set as a first guess with the  $M_{BFGS}$  method always leads to significant improvement of the model-data fit (see Figure S1) with RMSD reduction at the same level than as the median RMSD reduction for the  $M_{Genetic}$  method. This was not the case for the single-site optimisation (see Section 3.1.1). With a multiple-site optimisation the gradient-based scheme is less dependent on the first-guess, likely due to the smoothing

- 380 of the cost function as discussed above and therefore performing several random first-guess tests is less needed. Overall for the multiple sites optimisations the choice of the minimisation algorithm seems thus less crucial than for the single site cases. Nonetheless, the multi-site method conceals certain limitations. The optimisation may not perform well (i.e. not lead to a large minimisation of the cost function) if the different sites have very different behaviours in terms of the carbon and water cycle responses to climate forcing. This case points to the need to reconsider the PFT geographical description with possible
- 385 further regional split; it is suggested for TropEBF and C3 grass PFTs. Additionally, multiple-site optimisations, that requires large computing time, are more complicated to set up, with the need to have coherent observation errors between sites. We need to avoid that one or few sites dominate the cost function because of a too low observation error (measurement and model errors grouped in the R term) and thus prevent the optimisation to fit all the other sites.

**3.1.3 Benefit of multiple first guesses for a gradient-based approach**

- We now investigate more precisely quantitatively the benefit of using several first guess tests (16), especially for the gradient-based algorithm (see section 2.5). For a highly non-linear model the shape of *J* may be complex and the gradientbased algorithm can easily get stuck in a local minima (as mentioned above). Therefore it is useful or necessary to perform a set of independent assimilation runs starting with different first guesses. On the opposite side, a global search method is much less sensitive to the first-guess parameter values; specific choices related to the random exploration of the parameter space by the algorithm (i.e. the mutation rate, the number of chromosome and the number of iteration in the case of the
- Genetic Algorithm; see section 2.4.3) become crucial and the use of different first guesses is only an additional degree of freedom to explore the parameter space.

[revised manuscript text omitted]

all first-guess tests) encompass the true values ( $V_{cmax}$ ,  $c_{Tmin}$ ,  $C_{Tmax}$ , SLA,  $K_{GR}$ , MRoffset, MRslope, HRHc, HRHmin, Zdecomp,  $K_{albedo,veg}$ ,  $K_{rsoil}$ ).

- 445 As it was already outlined in Santaren et al. (2014), the differences obtained between the true and optimised parameter values are likely due to equifinality problems (i.e., multiple solutions achieving the same overall global minimum of the cost function) or the existence of local minimum-minima where the algorithm is trapped. These issues vary between parameters and are related to the sensitivity of the model outputs to each parameter, the prior parameter uncertainty and the level of error correlation between parameters. Given eExisting correlations and anti-correlations between the impacts of different
- 450 parameters<del>, it is not possible to retrieve</del> prevents from retrieving all of them with the chosen set of observations (daily means of NEE and LE) and the optimisation frameworks that are used.

Differences between the methods exist but are not systematic. There are a few parameters that perform well only with one of the methods, both for single and multiple sites optimisations. For example,  $c_{Topt}$ ,  $\tau_{teafinit}$  and  $Z_{decomp}$  are correctly estimated with only the gradient-based approach. On the contrary, the generic algorithm performs much better for parameters like LAImax, SLA,  $L_{age,crit}$ ,  $K_{LAIhappy}$ ,  $K_{rsoil}$ . A few parameters are not well estimated by both methods, like : the posterior values for  $F_{stressh}$ -and HRHb differ from the true value by 20% of the physical range- and  $Z_{crit,litter}$  by 30%, having the largest difference with respect to the true value. The estimated errors (Eq. 2) associated to these poorly retrieved parameters are relatively large and for  $F_{stressh}$  and HRHb they are highly correlated with other parameters; we should thus be very cautious when interpreting their value with real data optimisations. Overall, we observe that 19 parameters out of 28 are better estimated on average by the genetic algorithm (SGenetic, MGenetic) than by the gradient-based method (SBFGS, MBFGS). This is coherent with the results on

460 the genetic algorithm ( $S_{Genetic}$ ,  $M_{Genetic}$ ) than by the gradient-based method ( $S_{BFGS}$ ,  $M_{BFGS}$ ). This is coherent with the results on the fit to the data (section 3.1), indicating that the gradient method is more likely stuck in local minima than the genetic algorithm.

**3.2.2 Spread in parameter values across methods and first guess tests - real data experiments**

Using real data, figure 5 displays the mean posterior estimates of the TempDBF parameters across all first-guess tests for the four optimisation methods (SBFGS, SGenetic, MBFGS and MGenetic). For most of the parameters the mean optimised values are relatively similar between the gradient-based and the genetic algorithms, with some exceptions. 12 parameters (*G*s,slope, *c*Topt, *c*Tmin, *K*pheno,crit, *C*Tsen, LAImax, *K*LAIhappy, *Q*10, MRoffset, *K*GR, HRHc, *K*20) out of 28 show posterior differences between SGenetic and SBFGS methods that are lower than 5% of the physical range for each parameter, while 7 (*c*Tmax, *F*stressh, *K*wroot, SLA, τleafinit, HRHmin, *K*albedo,veg) show differences between 10 and 20%, whereas *K*rsoil goes up to 36%. For MGenetic and MBFGS methods, only 7 parameters are within the 5% variation range (*G*s,slope, *C*Tsen, LAImax, *K*z0 as for the single-site approaches, plus HRHa, *Z*decomp, *Z*crit,litter), and 14 parameters have differences over 10% (same list as for the single-site methods excluding *c*Tmax, plus *c*Topt, *c*Tmin, *K*pheno,crit, *C*Tsen, *Q*10, HRHc). However, few parameters demonstrate different behaviour between the real across the optimisation methods) correspond also to the most constraint ones in the pseudo-observation experiments (like *K*LAIhappy, MRoffset, LAImax).

For certain parameters ( $c_{\text{Topt}}$ , LAImax, SLA,  $L_{\text{age,crit}}$ ,  $K_{\text{LAIhappy}}$ ,  $K_{\text{GR}}$ ,  $K_{\text{rsoil}}$ ) we obtain significant differences in the estimated values between BFGS and GA, for both single-site and multi-site cases. Whereas the gradient-based methods looks for the optimal parameter set in the vicinity of the first-guess setup, the random search algorithm may jump to a completely different parameter state during one iteration. Given that with pseudo-data the GA manages to find the true values for these parameters much more precisely than the BFGS, we can speculate that the GA provides more optimal posterior estimates in

480

**the real data experiments as well.**

A secondAnother feature is that for the single-site and the multiple-site cases, both algorithms (BFGS and GA) lead for most parameters to a similar dispersion of the posterior estimates from the ensemble of 16 first-guesses (see figure 5). However, for some parameters ( $F_{\text{stressh}}$ , SLA, MRoffset, MRslope, HRHa, HRHb, HRHc, HRHmin,  $Z_{\text{crit,litter}}$ ,  $K_{\text{albedo,veg}}$ ,  $K_{\text{rsoil}}$ ) the GA method

- 485 gives, surprisingly, much higher distribution of the posterior values. It corresponds to the parameters that have also not been correctly estimated in the pseudo-observation tests (see section above) with estimated value outside the 10% range around the true value. Although not intuitive, the random sampling over the parameter space with the genetic algorithm could explain that for each first guess the GA method may end up exploring a larger domain of the parameter space and thus converge to more different parameter sets than the gradient-based method.
- 490 For each parameter, the dispersion obtained with the different first-guesses is lower for the multiple-site case compared to the single-site case, for both algorithms. On average the mean dispersion for the single-site approaches is 25% of the parameter range (with minimum and maximum spreads of 7.3% and 42%, respectively), and for the multiple-site approaches it is only 14% (between 3.4% and 32%). This reflects that with the multiple-site approach the shape of the cost function is more smooth due to the combination of different NEE/LE responses to the parameter variations (differences induce mainly
- by different climate, soil type and plant species) which in turns facilitate the convergence to the global minimum. For the single-site case, the shape of the cost function may differ significantly between sites leading to a larger parameter spread in response to multiple first guesses (compared to the multi-site case). This is specifically illustrated with the parameters  $K_{wroots}$ .  $C_{Tsen}$ ,  $Q_{10}$ , HRHa, HRHb and  $K_{z0}$ . The only exception is for  $K_{soilC}$ , a parameter that is specific to each site even in the multiplesite approach (see section 2.2).

**500 4 Summary and conclusion**

505

Throughout this study, we compared the performances of two algorithms for the optimisation of the ORCHIDEE land surface model parameters. The two algorithms belong to two different classes of method – L-BFGS-B algorithm <del>for theis a</del> gradient-based method<del>s (BFGS)</del> and the Genetic Algorithm (GA) for is a global random search method<del>s</del>. The two methods were used to optimise 28 parameters (16 being of which are PFT dependentdependent), using daily NEE and LE fluxes from 78 eddy covariance flux measurement sites. Both methods were run either independently at each site (single-site approach), running. fFor each configuration we ran 16 different

tests where in which the prior parameter values are selected randomly. The main findings are:

- For the single site (SS) optimisations-(SS), the random search algorithm (GA) results in lower values of the posterior cost function than the gradient-based method (BFGS) for nearly all sites.
- The difference between results derived from the GA and BFGS methods are smaller for multiple-site (MS) optimisations (MS)-than at single sites (SS)27. We suggest that smaller difference between GA and BFGS for MS optimisations is due to a smoothing of the cost function shape in the MS optimisations given with a greater number of observations are included in assimilation.
  - The GA results in a closer approximation of the true posterior parameter value than BFGS with the pseudo-observation tests, for both the SS and MS experiments.
    - When performing the same optimisation with multiple first guesses in parameter space (16 random first guesses), For the single site tests, the spread in posterior cost function minimum value when performing the same tests with multiple first guesses in parameter space (16 random first guesses) is much larger for the BFGS than the GA methods for SS optimisations, due to the higher likelihood that gradient based methods get stuck in local minima. The spread in posterior cost function value from 16 first guesses is
    - For the multiple-site tests, the spread in the cost function from 16 first guesses are closer for the MS optimisations, although still higher for the BFGS than the GA methods. Again, we suggest this is because the cost function is smoother in MS optimisations; therefore, the BFGS algorithm is less likely to get stuck in local minima.
- With-If using the BFGS method, our results suggest that performing the same tests with at least 5 differentseveral first guess parameters may ensure a reduction of the cost function that is comparable to the one obtained with random search GA method. This is important, because running a random search method may be computationally unfeasible for some modelling groups; therefore, the BFGS method can be used as a reliable alternative, provided a number of first guess parameters are used.
  - The GA results in a closer approximation of the true posterior parameter value than BFGS with the pseudoobservation tests, for both the SS and MS experiments.

The computing cost of the BFGS and GA algorithms were on average relatively similar, when considering all experiments, although slightly higher for the BFGS algorithm. With BFGS, the cost depends at which iteration the convergence criterion is met (see section 2.4.2). For the random search GA method, the value of the cost function may be further decreased by increasing the number of iterations (currently at 40). We choose a set up for the random search method that led to similar

535 computing cost than the gradient-based method, but this could be revised depending on the cost of a single-site model simulation (currently around 20-30 seconds with ORCHIDEE for one year using one standard processor depending on the number of output variables).

Most of the differences between the BFGS and GA algorithms are related to the shape of the cost function, in part controlled by the non-linearity of the model. Our results can thus be extrapolated to other land surface models, provided that they have similar complexity and level of non-linearity. With single siteSS optimisations, we advise to use a random search method\_\_\_\_

540

510

515

520

525

530

17

the Genetic Algorithm being just one possibility. If a gradient-based method was-is preferred, we strongly recommend performing at least several tests (5 or more) withusing different random first guess parameter values. Our study suggests that for the chosen model and inversion set up we should use at least 5 tests; however, this number will depend on the model complexity and particular set up of the DA experiment (e.g. type and number of observations included in the assimilation).

[revised manuscript text omitted]
 CO2 and H2O fluxes. Agricultural and Forest Meteorology, 148(10), 1467-1477, 2008.

Morris, M.D.: Factorial sampling plans for preliminary computational experiments. Technometrics, 33(2), 161-174, 1991.

Peaucelle, M., Bacour, C., Ciais, P., Peylin, P., Vuichard, N., Kuppel, S., Peñuelas, J.: Exploring plant functional traits
 variability with a terrestrial biosphere model, in review.

- Peylin, P., Bacour, C., MacBean, N., Leonard, S., Rayner, P., Kuppel, S., Koffi, E., Kane, A., Maignan, F., Chevallier, F. and Ciais, P.: A new stepwise carbon cycle data assimilation system using multiple data streams to constrain the simulated
  land surface carbon cycle. Geoscientific Model Development, 9(9), 3321, 2016.
- Pinnington, E. M., Casella, E., Dance, S. L., Lawless, A. S., Morison, J. I. L., Nichols, N. K., Wilkinson, M. and Quaife, T.
   L.: Investigating the role of prior and observation error correlations in improving a model forecast of forest carbon balance using Four Dimensional Variational data assimilation. Agricultural and Forest Meteorology, 228-229, 299-314, doi:10.1016/j.agrformet.2016.07.006, 2016.
- Post, H., Vrugt, J.A., Fox, A., Vereecken, H. and Hendricks Franssen, H.J.: Estimation of Community Land Model parameters for an improved assessment of net carbon fluxes at European sites. Journal of Geophysical Research:
   Biogeosciences, 122(3), 661-689, 2017.
- Raoult, N. M., Jupp, T. E., Cox, P. M., and Luke, C. M.: Land-surface parameter optimisation using data assimilation techniques: the adJULES system V1.0, Geosci. Model Dev., 9, 2833-2852, doi:10.5194/gmd-9-2833-2016, 2016.
  Reichstein, M.: Inverse modeling of seasonal drought effects on canopy CO2 / H2O exchange in three Mediterranean ecosystems, J. Geophys. Res., 108, 4726, doi:10.1029/2003JD003430, 2003.
- Ricciuto, D.M., King, A.W., Dragoni, D. and Post, W.M.: Parameter and prediction uncertainty in an optimized terrestrial carbon cycle model: Effects of constraining variables and data record length. Journal of Geophysical Research: Biogeosciences, 116(G1), 2011.

Richardson, A.D., Williams, M., Hollinger, D.Y., Moore, D.J., Dail, D.B., Davidson, E.A., Scott, N.A., Evans, R.S., Hughes,
H., Lee, J.T. and Rodrigues, C.: Estimating parameters of a forest ecosystem C model with measurements of stocks and
fluxes as joint constraints. Oecologia, 164(1), 25-40, 2010.

[revised manuscript text omitted]

\* TropEBF — tropical evergreen broadleaf forest; TempENF — temperate evergreen needleleaf Forest; TempEBF — temperate evergreen broadleaf forest; TempDBF — temperate deciduous broadleaf forest; BorENF — boreal evergreen needleleaf forest; BorDBF — boreal deciduous broadleaf forest; C3 grass — C3 grassland.